# An improved Bayesian inversion to estimate daily NO$_x$ emissions of Paris from TROPOMI NO$_2$ observations between 2018-2023

Alba Mols[1], Klaas Folkert Boersma[1,2], Hugo Denier van der Gon[3], and Maarten Krol[1,4]

[1]Wageningen University, Meteorological and Air Quality department, Wageningen, the Netherlands
[2]Royal Netherlands Meteorological Institute (KNMI), De Bilt, the Netherlands
[3]Department of Climate, Air and Sustainability, TNO, Utrecht, the Netherlands
[4]Institute for Marine and Atmospheric Research Utrecht (IMAU), Utrecht University, Utrecht, the Netherlands

**Correspondence:** Folkert Boersma (folkert.boersma@wur.nl)

**Abstract.** We present a comprehensive quantification of daily NO$_x$ emissions from Paris using an inverse analysis of tropospheric NO$_2$ columns measured by the Tropospheric Monitoring Instrument (TROPOMI) over a 5-year period (May 2018 - August 2023). Our analysis leverages a superposition column model that captures the relationship between the increase in NO$_2$ with distance over an urban source region to underlying NO$_x$ emissions, accounting for chemical transformations and wind in
the urban boundary layer. To evaluate the robustness of the superposition column model, we tested it against high-resolution (300 m) Large Eddy Simulations (LES) using MicroHH, a computational fluid dynamics model, with atmospheric chemistry, confirming that the model's simplifying assumptions introduce uncertainties below 10%. Building on this foundation, we develop a new Bayesian inversion method that incorporates prior knowledge on NO$_x$ emissions and lifetimes and accounts for model and prior uncertainties. Compared to a previous look-up table approach, which relied on least-squares minimization
without prior constraints, the Bayesian method demonstrated superior performance. In controlled tests, it reproduced known NO$_x$ emissions within 5%. Applying Bayesian inversion to TROPOMI data in Paris, we observed a significant reduction in NO$_x$ emissions from 44 mol s$^{-1}$ in 2018 to 32 mol s$^{-1}$ in 2023, representing a 27% decrease. This decline exceeds the 12% reduction predicted by the TNO-MACC-III bottom-up inventory, indicating limited accuracy of current inventories. Seasonal analysis revealed higher posterior emissions in winter, possibly highlighting the role of residential heating or vehicle cold
starts, which may be underrepresented in bottom-up estimates. Our improved Bayesian framework delivers accurate NO$_x$ emission estimates that align well with independent data sets. This approach provides a valuable tool for monitoring urban NO$_x$ emissions and assessing the efficacy of air quality policies.

## 1 Introduction

Nitrogen oxides (NO$_x$=NO+NO$_2$) are major air pollutants which are central to the chemistry of the troposphere, and which
have negative impacts on human health and the environment (e.g. Boningari and Smirniotis (2016)). In urban regions NO$_x$ is mainly emitted to the atmosphere as a result of the burning of fossil fuels, particularly in combustion engines. In the EU, the largest contributor to NO$_x$ emissions is the transport sector (40%), followed by energy production and distribution (16%), and the commercial, institutional and households sectors (15%) (EEA, 2019). At daytime, nitrogen oxides are short-lived, on

the order of 1-12 hours (Stavrakou et al., 2013), because $NO_2$ is quickly oxidized by reaction with the hydroxyl radical (OH) to nitric acid ($HNO_3$). Due to its high water solubility, $HNO_3$ is efficiently removed from the atmosphere, primarily through precipitation and direct deposition onto surfaces (Seinfeld and Pandis, 2016).

Besides being a toxic gas itself, $NO_2$ also has secondary effects via its contribution to photochemical ozone production (Seinfeld and Pandis, 2016; Jacob, 1999; Visser et al., 2019), its influence on the formation of aerosols (Yan et al., 2020), and its contribution to eutrophication via the deposition of $HNO_3$ to ecosystems (e.g. Vitousek et al. (1997); Erisman and Draaijers (1995)).

To reduce the negative effects of $NO_x$, the EU maintains a limit value for average annual surface air $NO_2$ concentrations of 40 $\mu$g m$^{-3}$. Currently, air quality in most European cities complies (EEA, 2022). Nevertheless, $NO_x$ pollution remains a significant health concern for Europeans, especially in urban areas, as daily guidelines set by the World Health Organisation (WHO) of 25 $\mu$g m$^{-3}$ (WHO, 2021) continue to be frequently exceeded (EEA, 2022).

Monitoring of $NO_x$ emissions typically relies on bottom-up inventories, which are uncertain due to their reliance on emission factors, extrapolations and activity assumptions. Uncertainties in bottom-up emissions vary with location, and are estimated to be typically more than 30% (Kuenen et al., 2014). Satellite measurements of $NO_2$ offer a useful tool for top-down inverse modelling of $NO_x$ emissions, providing more insights into $NO_x$ sources and distributions. However, inverse modelling is also subject to assumptions, such as uncertainties of $NO_x$ lifetimes in the lower atmosphere (Stavrakou et al., 2013), which can significantly influence the accuracy of top-down emission flux estimates.

Research and refinement of inversion methods for estimating $NO_x$ emissions and lifetimes are crucial, especially for initiatives like the Copernicus CO2M mission (Sierk et al., 2021), which will utilize $NO_2$ plumes to enhance $CO_2$ monitoring by pinpointing emissions more accurately. Several studies have quantified $NO_x$ emissions based on satellite $NO_2$ retrievals, by analyzing downwind plumes of $NO_2$ from large sources, using inverse modeling computations with atmospheric chemical transport models (CTMs) (e.g. Brioude et al. (2013); Cheng et al. (2021); Kurokawa et al. (2009); Krol et al. (2024); Zhu et al. (2022)). However, because CTMs can have accessibility issues and require substantial computational resources, alternative approaches that do not depend on CTMs have been developed and utilized to estimate $NO_x$ emissions and lifetimes (e.g. de Foy et al. (2014), Beirle et al. (2011)).

Beirle et al. (2011) first presented a method to infer $NO_x$ emissions from strong isolated sources, by averaging satellite $NO_2$ plumes with similar wind direction. Building upon this concept, Lorente et al. (2019) presented a simple superposition column model that uses $NO_2$ retrievals over Paris of the TROPOspheric Monitoring Instrument (TROPOMI), combined with domain-average information about the wind speed, wind direction and OH concentrations in the boundary layer, to estimate urban $NO_x$ emissions and lifetimes without the need for complex inverse modelling computations. This approach allows for day-to-day emission estimates under cloud-free conditions, offering the potential for continuous emission estimations over a long period of time. Zhang et al. (2022) expanded this model framework to estimate the $NO_x$ and $CO_2$ emissions originating from Wuhan, introducing modifications to the method that included considering chemical decay of upwind background $NO_2$ flowing into the city. Inverse modelling approaches derived from the method of Beirle et al. (2011), as exemplified by Lorente et al. (2019) and subsequent studies (e.g. Zhang et al. (2022), Goldberg et al. (2022), de Foy and Schauer (2022), Lange et al. (2022), Liu et al.

(2022), Rey-Pommier et al. (2022)), inherently simplify the effects of atmospheric dynamics and chemistry. These methods have nonetheless been evaluated using synthetic data, with studies such as de Foy et al. (2014) and Liu et al. (2022) showing that inferred $NO_x$ emissions and lifetimes remain broadly consistent with known model input. In a complementary approach, Zhu et al. (2022) inferred long-term changes in $NO_x$ lifetime from decadal OMI $NO_2$ columns, using machine learning to relate $NO_2$ columns urban OH concentrations. Simplifications arise from assumptions that spatially and temporally varying wind speeds, $NO_x/NO_2$ ratios, and $NO_x$ lifetimes may be taken as constant throughout the inversion domain, whereas in reality there may be substantial temporal and spatial fluctuations in these parameters, especially near the edges of plumes (Hakkarainen et al., 2024; Krol et al., 2024; Meier et al., 2024; Valin et al., 2013; Vilà-Guerau de Arellano et al., 2004). We therefore address the following research questions:

1. To what extent is the forward superposition model capable of simulating realistic $NO_2$ concentrations, despite simplifications on domain-average wind speed, $NO_x/NO_2$ ratios, and $NO_x$ lifetimes?

We revisit the methodology introduced by Lorente et al. (2019) and perform an Observing System Simulation Experiment (OSSE). We generate synthetic satellite $NO_2$ observations using two Large Eddy Simulation (LES) experiments for a hypothetical city to investigate realistic chemical variations that occur in urban plumes. We assess to what extent the simple column model of Lorente et al. (2019) is capable of appropriately capturing $NO_2$ increases along with the wind over a city despite these simplifications. Next, we move on to the inversion of $NO_x$ emissions, and pose the question:

2. Can a Bayesian inversion method that weighs prior information, forward model uncertainty, and observational uncertainties improve estimates of $NO_x$ emissions from TROPOMI $NO_2$ plumes relative to a method that does not account for constraints imposed by prior knowledge?

We propose and evaluate a new more formal Bayesian inversion method, incorporating prior knowledge on $NO_x$ emissions and $NO_x$ lifetime and observations of $NO_2$ and their uncertainties. In Sections 2 and 3, we use the synthetic observations sampled from simulations with a high-resolution computational fluid dynamics model that resolves large-scale turbulence and atmospheric chemistry (MicroHH (Van Heerwaarden et al., 2017)) to evaluate the forward superposition model and inferred emissions. Then in Section 4, to demonstrate the applicability of this method, we infer a 5-year timeseries of $NO_x$ emissions for Paris on individual clear-sky days between June 2018 - August 2023 using improved TROPOMI $NO_2$ V2.4.0 retrievals. This retrieval product is based on high-resolution (0.125°) surface albedo information from the DLER database (Tilstra et al., 2023) and high-resolution (0.1°) a priori $NO_2$ profiles from CAMS (Douros et al., 2022). TROPOMI $NO_2$ retrievals based on high-resolution input data have been shown to capture $NO_2$ gradients well (Lange et al., 2024). The inversion illustrates the potential of the method and allows us to identify trends and patterns in the $NO_x$ emissions of Paris, including seasonal and weekly emission cycles, and to assess the effectiveness of pollution reduction efforts. We conclude with an evaluation of our top-down $NO_x$ estimates with an independent bottom-up inventory of $NO_x$ emissions of Paris.

## 2 Forward superposition column model

### 2.1 Model setup

The superposition column model first presented by Lorente et al. (2019) calculates $NO_2$ columns by superimposing $NO_x$ emissions along the wind within a region. The region (e.g. a city) that is analysed is divided into line cells that encapsulate the entire source region perpendicular to the wind direction (Fig. 1, (Beirle et al., 2011)). For each cell $i$ between, within, and beyond the city length, the contribution to the $NO_2$ line density downwind to the cell is calculated using a simple column model:

$$
\begin{aligned}
f_i(x) &= 0 & \text{for } x < x_i \\
f_i(x) &= \frac{E_i}{k}(1 - e^{-k(x-x_i)/u}) \times \frac{[NO_2]}{[NO_x]} & \text{for } x = x_i \\
f_i(x) &= \frac{E_i}{k}(1 - e^{-kL/u}) \times e^{-k(x-x_i)/u} \times \frac{[NO_2]}{[NO_x]} & \text{for } x > x_i
\end{aligned}
\tag{1}
$$

Where $f_i$ is the contribution of the emissions in cell $i$ to the $NO_2$ line density at $x$ (mol cm$^{-1}$), $E(x_i)$ represents the $NO_x$ emissions from cell $x_i$ (mol cm$^{-1}$ s$^{-1}$), $L$ is the length of each line cell (m) and $u$ is the effective wind speed at which $NO_x$ is transported (m s$^{-1}$). This effective wind speed is determined by weighing the vertical wind speed profile by the the vertical $NO_2$ density profile, as described in Lorente et al. (2019). The scaling with the $[NO_2]/[NO_x]$ ratio is required because a fraction of $NO_x$ is present as $NO_2$, and TROPOMI measures the $NO_2$ columns. $k$ is the rate constant of the chemical loss of $NO_x$ during daytime (s$^{-1}$): $k = \frac{k'[OH]}{[NOx]/[NO2]}$, using the reaction rate constant $k'$ of $1.1 \cdot 10^{-11}$ cm$^3$ molecule$^{-1}$ s$^{-1}$ for the OH + $NO_2$ + M reaction at surface pressure for 288K (Burkholder et al., 2020). PAN formation is not explicitly considered in this framework, as it is a reversible $NO_x$ reservoir rather than a permanent sink (e.g., (Fischer et al., 2014)). In the warm, VOC-limited conditions typical of central Paris (e.g., Johnson et al. (2024)), PAN decomposes rapidly and contributes little to net $NO_x$ loss. The dominant $NO_2$ sink under these conditions is oxidation to $HNO_3$, with some additional loss to organic nitrates ($RONO_2$).

The contributions to the line density from each cell are added to the background $NO_2$ concentration ($b$) to find the overall $NO_2$ line density at each distance $x$ along with the wind:

$$
F(x) = \sum_{i=1}^{n} f_i(x) + b
\tag{2}
$$

Following this model formulation, the $NO_2$ that accumulates over the city $F(x)$ depends on the spatial pattern of emissions $E(x)$ within the city and is affected by the chemical loss and the wind speed over the city, as discussed extensively in the studies by Lorente et al. (2019) and Zhang et al. (2022). The superposition column model defined by Eqs. (1) and (2) implies that prior knowledge is required on oxidation chemistry (OH concentration and $NO_x$:$NO_2$ ratio) within the urban boundary layer, and that the background $b$ represents the spatially invariant free tropospheric $NO_2$ contribution to the line density.

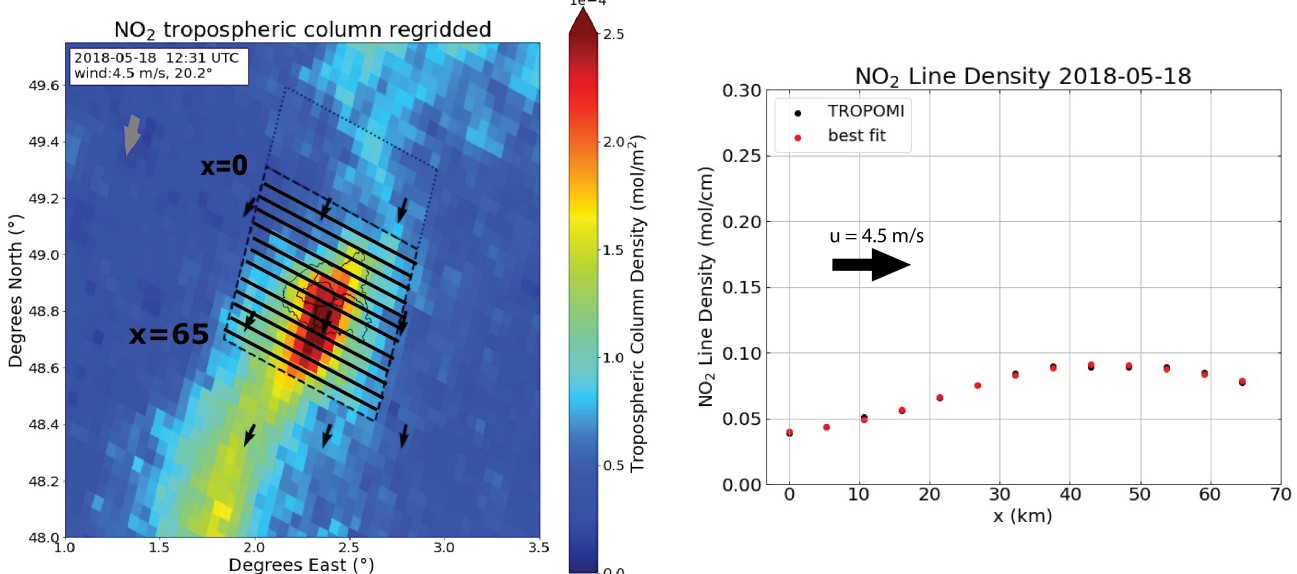

**Figure 1.** Left: the study area of wider Paris and illustration of the line density method. The black solid lines indicate the different line cells. The black arrows indicate the wind speed (from the north-east). The dotted grey box to the north of Paris represents the background area. Right: $NO_2$ line densities at each distance $x$, calculated from the TROPOMI retrieval on 18 May 2018.

## 2.2 Comparison superposition model to synthetic $NO_2$ satellite observations with MicroHH

To generate synthetic satellite observations of $NO_2$, we use MicroHH, a direct numerical simulation (DNS) and large-eddy simulation (LES) model (Van Heerwaarden et al., 2017), which has recently been extended to include an atmospheric chemistry

module based on the Kinetic Pre-Processor package (KPP) (Krol et al., 2024). We set up a horizontal model domain of 50 km (North-South) $\times$ 150 km (East-West) $\times$ 4 km (vertical), with a horizontal resolution of 300 m and a vertical resolution of 100 m. At the upper end of the model domain, a buffer zone of 750 m serves to damp gravity waves (Van Heerwaarden et al., 2017). For temperature, humidity, and momentum, circular boundary conditions were used. To avoid re-entering of emissions from the city source, we employed free outflow conditions for tracers (Ražnjević et al., 2022). More information about the MicroHH

setup and initial conditions used can be found in section 1 of the Supplementary Material.

We conduct two MicroHH simulations over a hypothetical city ('symcity') to assess the capability of the forward line density model (Eq. 1,2) to realistically simulate line densities over a city, despite its simplifications (a spatio-temporally constant windspeed, $NO_x$:$NO_2$ ratio, and $NO_x$-lifetime). Our simulated city, 30 km $\times$ 30 km in size, is positioned on the west side of the model domain, which is dominated by a westerly flow. $NO_x$ is emitted within the city as NO, and gradually transformed to $NO_2$

by reaction with ozone. The NO emissions are spatially distributed over the city in a Gaussian pattern, so that the emissions are much larger in the center. 68% of the emissions lie within a radius of 7.5 km around the city center, and 95% within 15 km.

The simulation is performed between 6:00 and 18:00 hrs local time. Photolysis representative for the city of Riyadh is used in the simulation. We simulate two different scenario's: scenario 1 is a Spring case (photolysis of April 15th) which has high NO emissions of 195.7 mol s$^{-1}$, and a high wind speed of 6 m s$^{-1}$. Scenario 2 is a winter case (photolysis of December 15th) and has lower NO emissions of 58.7 mol s$^{-1}$ and a wind speed of 2 m s$^{-1}$. Details of the two cases are summarized in Table 1. Figure 2 displays the MicroHH tropospheric NO$_2$ columns between 0-4 km height at 13:00 hrs, close to the approximate TROPOMI overpass time of 13:30 hrs. Since the emissions are smoothly distributed over the city, the irregularities in the simulated NO$_2$ columns, visible in the upper panels, are caused by the combined effects of atmospheric turbulence and chemistry. The lower panel shows the columns averaged to a 3 km $\times$ 3 km resolution, more similar to the spatial resolution of the TROPOMI NO$_2$ retrievals.

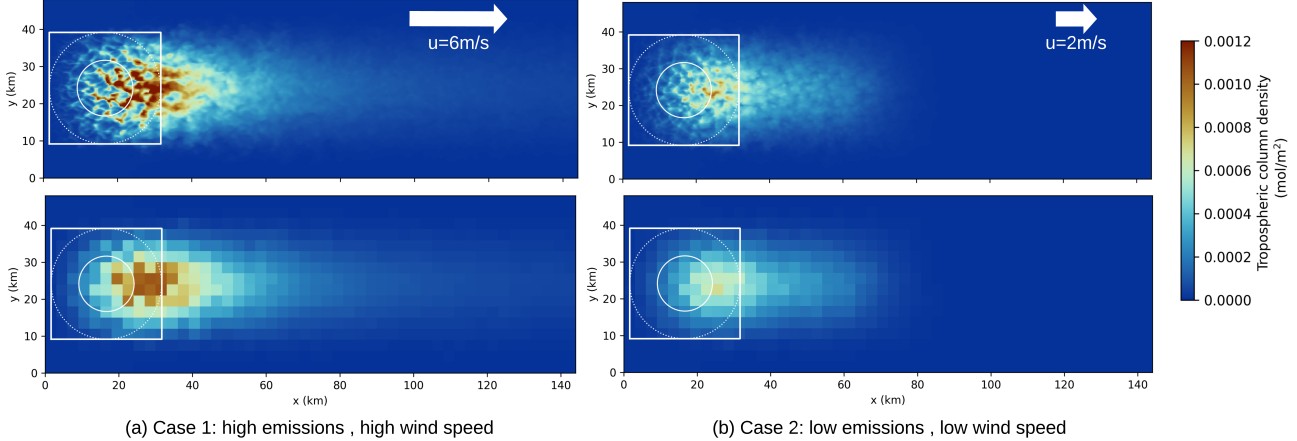

(a) Case 1: high emissions , high wind speed     (b) Case 2: low emissions , low wind speed

**Figure 2.** Tropospheric NO$_2$ columns of symcity simulated by MicroHH for a Spring atmosphere with high NO$_x$ emissions and strong wind (left panels), and a Winter atmosphere with low NO$_x$ emissions and weak wind (right panels). 68% of the NO$_x$ emissions occur within the smallest circle, and 95% within the bigger circle. The upper panels show the NO$_2$ columns at the MicroHH spatial resolution of 300 m, and the lower panels show the same MicroHH simulations regridded to 3 km resolution.

Building upon the 3 km $\times$ 3 km resampled simulations, we computed NO$_2$ line densities across symcity, depicted as black dots in Fig. 3c and 3f. The simulated NO$_2$ line densities over the city exhibit a tilted S-shaped pattern, similar as the observed line densities over Paris reported in Lorente et al. (2019). This pattern is a consequence of the dynamic interplay between wind and the Gaussian emission distribution across the city, with maximal emissions concentrated at the city center. NO$_2$ columns are very low upwind of the city, and the NO$_2$ line densities increase once the emitted NO is converted into NO$_2$ via the NO+O$_3$ reaction in the boundary layer. For the Spring case, with 6 m s$^{-1}$ wind speed, the NO$_2$ line density peaks downwind of the city, reflecting the rapid transport of NO$_2$ beyond the city limits. The Winter case shows the NO$_2$ line density peaks over the city, at about 25 km, reflecting the lower wind speed in that simulation.

We evaluate the forward superposition model (based on Eq. (1) and Eq. (2)) by comparison to line densities we directly obtained from MicroHH. For the forward model, we use the $NO_x$ emissions (magnitude and spatial distribution) and other model input parameters (the average $NO_x$ lifetime, wind speed, and the average $NO_x/NO_2$ ratio (Eq. 1)) from the symcity test (these average values are listed in Table 1). We try to determine these parameters as closely as possible to data we would obtain from CAMS (Copernicus Atmosphere Monitoring Service), to ensure a realistic representation of conditions encountered in a TROPOMI inversion scenario (Lorente et al., 2019). The high resolution of the MicroHH model enables us to discern the implications of these simplifications. We calculate the parameter values from the MicroHH output of 12:00 hrs local time, as the observed $NO_2$ columns depend on the conditions of preceding time steps. In the MicroHH simulation, the $NO_x$ lifetime ranges between 1-6 hours (2-9 h for case 2) within the city domain (Fig. 3a,d). The $NO_x$ lifetime is especially short in the downwind part of the plume, reflecting high OH concentrations in the urban plume (there were also substantial hydrocarbon emissions from the city, which leads to $O_3$ formation and consequently enhanced OH). The $NO_x/NO_2$ ratio ranges between 1.2-1.7 (1.2-1.6 for case 2) over the city domain (Fig. 3b,e). For the Winter case (case 2) $O_3$ and OH concentrations are slightly lower than for the Spring case (case 1). This leads to a slightly longer $NO_x$ lifetime, and a lower $NO_x/NO_2$ ratio than for the Spring case. In the MicroHH simulations, the wind speed remains relatively constant across the entire domain. The domain average wind speed, weighted vertically by the vertical $NO_2$ concentration is around 6 m/s for Spring and 2 m/s for Winter. Our superposition model requires effective values of wind, $NO_x/NO_2$ and $NO_x$ lifetime as input parameters. Simulating $NO_2$ line densities over the city using the superposition model with the domain averaged input parameters from MicroHH yields the values shown as red dots in Fig. 3c,f. We see that despite simplifications, the simulated line densities from the superposition model closely match those from the MicroHH simulation. The agreement between the superposition and MicroHH $NO_2$ line densities allows us to estimate the superposition forward model error. We estimated the forward model error as the average absolute deviation for the 10 line density values along with the wind. It amounts to 6.5% from the average MicroHH line density for case 1 and 6.3% for case 2. This suggests that, despite simplifications, the superposition model is effective and provides realistic $NO_2$ line densities based on city-domain averaged $NO_x$ emissions, lifetime and $NO_x/NO_2$ ratio and wind speeds, at least at the spatial resolution of TROPOMI $NO_2$ observations.

**Table 1.** The range of parameter values within the city domain for the two cases of the MicroHH simulations for a Riyadh-like city, at 13:00, and the domain-averaged parameters used in the forward model simulations. The wind speed used in the forward model is the height-averaged wind speed weighted by the vertical $NO_2$ distribution. The range displayed for MicroHH is the city-domain averaged wind speed at 100 m height and 3 km height.

| | Photolysis | | Total $E_{NO_x}$ (mol s$^{-1}$) | Wind speed (ms$^{-1}$) | $NO_x$ lifetime (h) | $NO_x/NO_2$ ratio | Line densities |
|---|---|---|---|---|---|---|---|
| Case 1 | 15 Apr | MicroHH | 195.7 | 4 (100 m) - 8 (3 km) | 2.6±1.2 | 1.5±0.08 | Black dots in Fig. 3c |
| | | Forward model | | 5.7 | 2.21 | 1.48 | Red dots in Fig. 3c |
| Case 2 | 15 Dec | MicroHH | 58.7 | 2 (100 m) - 3 (3 km) | 3.4±1.8 | 1.4±0.08 | Black dots in Fig. 3f |
| | | Forward model | | 2.3 | 2.87 | 1.4 | Red dots in Fig. 3f |

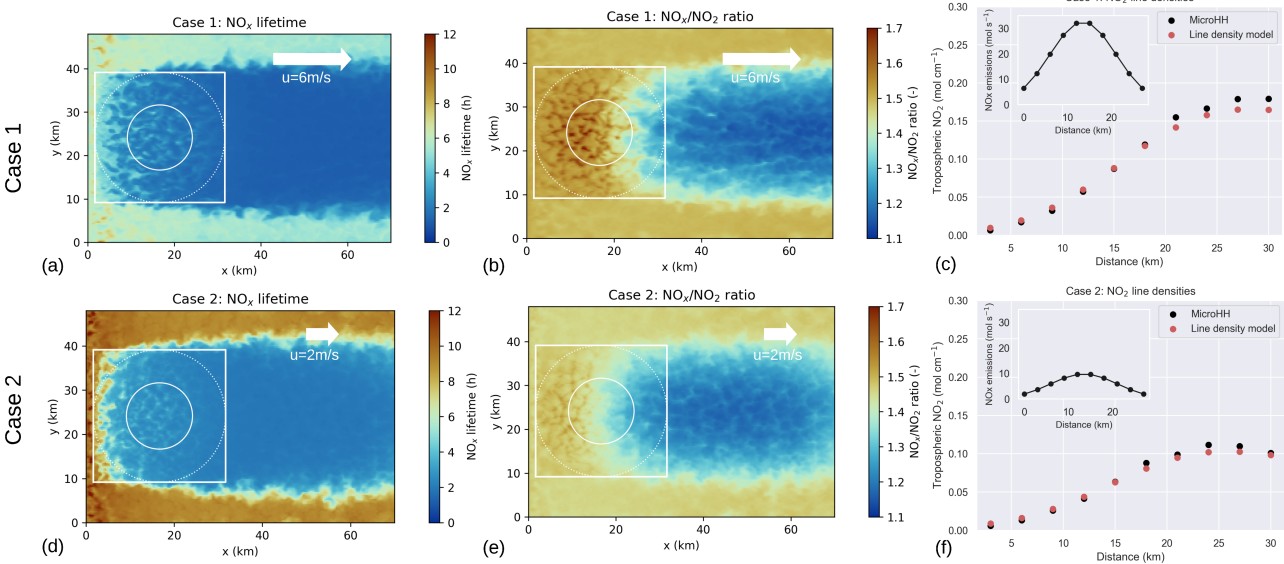

**Figure 3.** NO$_x$ lifetimes and NO$_x$/NO$_2$ ratios for the two cases as simulated by MicroHH over the entire domain. Simulated line densities over symcity of MicroHH and the line density model (c,f). The emission profile over the city is displayed in the small panel. The city-domain averaged conditions are given in the white box. Figures are for 13:00 hrs local time.

## 3 Inversion of NO$_x$ emissions and lifetime

We now assess the ability of the inverse superposition model to estimate known input NO$_x$ emissions and lifetimes based on the
175 NO$_2$ line densities as simulated by MicroHH, again using the two MicroHH simulations regridded to a TROPOMI resolution.

### 3.1 Inversion methods

Since TROPOMI measures NO$_2$ columns, we need to estimate the NO$_x$ emissions using auxiliary and a priori knowledge
of wind, chemical regime and emissions. In Lorente et al. (2019) a simple inversion method is used for this: the forward
model (Eq. 1, 2) is fitted to the observed NO$_2$ line densities by minimizing the sum of the squares of the residuals, using a
180 pre-calculated look-up table with a large number of NO$_2$ line densities corresponding to combinations of NO$_x$ emissions and
NO$_x$ lifetimes, which are allowed to vary by $\pm 50\%$ from their prior values. No formal weight is assigned to prior knowledge
regarding NO$_x$ emissions and lifetimes. The optimal solution is determined by the NO$_x$ emissions and lifetimes that result
in the lowest residuals between observed and pre-calculated line densies. The optimal solution (with the lowest residuals)
according to the procedure in Lorente et al. (2019) may therefore include some estimates of NO$_x$ emissions and lifetimes that
are unrealistic, e.g. with large emissions accompanied by large OH.

Here we propose a new, more formal inversion method that uses the minimization of the Bayesian cost function, taking into account knowledge on uncertainties of the prior emissions and lifetime, uncertainties in the forward model (see section 2.2), and uncertainties in the measured line densities:

$$J(x) = \frac{(x - x_a)^2}{\sigma_A^2} + \frac{(F(x) - y)^2}{\sigma_O^2}$$ (3)

Where $x$ is the state vector, including all the terms that are fitted ($k$, and the emissions from each line cell). The cost function is minimized by finding the solution of $dJ/dx = 0$. The cost function comprises two different terms. The first term is the deviation of the prior estimate ($x_a$) of the state from the actual state ($x$). The second term is the deviation of the calculated line densities ($F(x)$) from the measured line densities ($y$), given the solution for the state $x$. Both terms are weighted by their uncertainties. $\sigma_A$ represents the uncertainty in the prior, and $\sigma_O$ represents the combined measurement uncertainty and the uncertainty in

the model representation of the system. With this Bayesian inversion method, we take into account not just the observations but also our prior knowledge. This prevents the model from excessively conforming to the observed data ("overfitting"), which is problematic in the basic inversion method. The main differences between the least-squares inversion method from Lorente et al. (2019) and the Bayesian inversion approach are displayed in Table 2.

**Table 2.** Differences between the least-squares and the Bayesian inversion method in inferring the NO$_x$ emissions and lifetimes from the NO$_2$ line densities.

| | (1) least-squares inversion | | (2) Bayesian inversion | |
|---|---|---|---|---|
| Cost function | $J(x) = (F(x) - y)^2$ | (4) | $J(x) = \frac{(x - x_a)^2}{\sigma_A^2} + \frac{(F(x) - y)^2}{\sigma_O^2}$ | (5) |
| Condition state parameters | $\tau \pm 0.5\tau$ | | $\tau \pm \sigma_{A,\tau}$ | |
| | $E \pm \infty$ | | $E \pm \sigma_{A,E}$ | |

## 3.2 Symcity emission inversion

We now apply the two inversion methods to infer the NO$_x$ emissions and lifetimes of symcity based on the NO$_2$ columns from MicroHH, for both the high and low emission scenario. First, we assume zero observational error, and only uncertainty in the model representation of the system. We use a $\sigma_O$ of 6%, representing the model representation uncertainty that we determined in section 2.2. We use a prior lifetime uncertainty $\sigma_{a,\tau}$ of 30%, and a prior emission uncertainty $\sigma_{a,E}$ of 50%. First, we assume known prior conditions, so the emission profiles over the city and city-average lifetimes from MicroHH are used as the prior for

the Bayesian model. The results of the inversions are presented in Table 3. The Bayesian inversion method yields emissions that closely match those input into MicroHH (within 2%). The least-squares inversion method slightly underestimates emissions (4%) and overestimates lifetimes for case 1, and vice versa for case 2 (16% emission overestimation) highlighting overfitting

issues inherent in the least-squares inversion approach. Although the Root Mean Square Error (RMSE) is smaller when using the least-squares inversion method, there are slight discrepancies in the inferred emissions and lifetimes due to this overfitting.

**Table 3.** Comparison of $E_{NO_x,tot}$, $\tau_{NO_x}$, and RMSE values for different inversion methods: MicroHH, least-squares inversion, and Bayesian inversion.

| | Case 1 | | | Case 2 | | |
|---|---|---|---|---|---|---|
| | $E_{NO_x,tot}$ (mol s$^{-1}$) | $\tau_{NO_x}$ (h) | RMSE (mol s$^{-1}$) | $E_{NO_x,tot}$ (mol s$^{-1}$) | $\tau_{NO_x}$ (h) | RMSE (mol s$^{-1}$) |
| MicroHH | 195.7 | 2.21 | | 58.7 | 2.87 | |
| Least-squares inversion | 188 | 3.31 | 0.002 | 69.9 | 2.26 | 0.0016 |
| Bayesian inversion | 196.7 | 2.77 | 0.003 | 57.9 | 3.09 | 0.0021 |

This first inversion was performed for the idealized scenario, with zero observational uncertainty. To enhance the similarity of the NO$_2$ columns in MicroHH to what TROPOMI would observe, we introduced uncertainty on top of the MicroHH-simulated NO$_2$ columns. We prescribe this uncertainty as:

$$\sigma = 0.4 \cdot 10^{15} + 0.2 \cdot N_v \ \text{(molecules cm}^{-2}) \tag{6}$$

Where $N_v$ is the NO$_2$ column. The first part represents random uncertainty in TROPOMI measurements (originating from measurement noise in the satellite level-1 data), while the second part accounts for systematic uncertainty in estimating the Air Mass Factor (AMF), for instance caused by to uncertainties in albedo estimates (Van Geffen et al., 2022; Riess et al., 2023). The systematic part is correlated between adjacent cells, with a Gaussian-like shape between adjacent cells with a spatial correlation length (where the correlation falls to 1/e) of 7km (e.g. Rijsdijk et al. (2025)). We perform 1000 inversions with different random uncertainty, drawing them from a normal distribution with the standard deviation defined in Eq. 6.

The results are displayed in Fig. 4. Here we show the mean emission estimates for all 1000 runs with uncertainty on the NO$_2$ columns, for the true MicroHH emissions, the prior estimate, the Bayesian emission inversion and the Least-squares inversion. We do this for both case 1, the Spring case with high emissions (Fig. 4a) and case 2, the Winter case with lower emissions (Fig. 4b). In this more realistic setup, which uses realistic TROPOMI uncertainties instead of the idealized inversion of Table 3, emissions are more frequently overestimated in both MicroHH cases when using the least-squares inversion method than when using the Bayesian method.

We also investigate the extent to which NO$_x$ emissions and lifetime can be independently and simultaneously reproduced. Figure 4b,d shows the correlation between the errors in the inferred lifetime and emissions. The observed errors in the inferred NO$_x$ emissions and lifetimes exhibit substantial correlation. An overestimation in emissions is consistently accompanied by an underestimation in lifetime and vice versa. This relationship is in line with the fundamental equation of the superposition column model (Eq. 2), where both elevated emissions and extended lifetimes contribute to increased line density values. These inversions of the emissions from MicroHH provide evidence and quantitative insights into the strength of this correlation. The figures demonstrate a nonlinear relationship between the errors in inferred NO$_x$ emissions and lifetimes, where the cross-

correlation appears strongest when the lifetime is underestimated. This points to a logarithmic cross-correlation between the error in $NO_x$ lifetimes and emissions, suggesting that small deviations in lifetime have a more substantial impact on emissions when the lifetime is underestimated.

The deviations in the inferred emissions are generally larger for the least-squares inversion method. Figures 4b,d clearly show that the lifetimes tend to be estimated at either the -50% or +50% limit that is restricted in the fit. Because of this, the emissions are also estimated in two modes, where one is close to the true value and in the other case, emissions are overestimated. For the Bayesian model, the fit is kept in check by the prior estimate, preventing overfitting, and leading more often to the right emission estimate. For the Bayesian inversion, the median error in the emission estimate of case 1 (case 2 in brackets) is -0.7% (-5.1%) and the standard deviation is 6.9% (11.3%). For the least-squares inversion, the median is 14% (26%) and the standard deviation is 22% (34%). Regarding the lifetime, the median error in the lifetime estimate of case 1 (case 2 in brackets) is 32% (13%) for the Bayesian inversion and the standard deviation is 31% (61%). For the least-squares inversion, the median is 2% (-19%) and the standard deviation is 46% (37%). To investigate the sensitivity of our results to deviations in the prior, we conducted an additional test. We repeated the OSSEs for both Symcity cases 50 times, introducing a $\pm 20\%$ deviation in either the prior lifetime or emissions. These sensitivity tests show that increasing or decreasing the prior emissions by 20% results in a posterior bias of no more than 6% compared to the case with a known prior. Detailed results can be found in section 2 of the Supplementary Material.

We showed here that the strong correlation between errors in estimated $NO_x$ emissions and lifetimes makes their independent estimation difficult. Some studies estimate $NO_x$ lifetimes by analyzing the exponential decay of the $NO_2$ plume downwind of a city (e.g. Beirle et al. (2011); de Foy et al. (2014); Liu et al. (2022). While this e-folding distance approach can provide additional constraints on the $NO_x$ lifetime compared to our method, which relies solely on the enhancement of $NO_2$ over the city, it does not account for variations in photochemistry between the urban area and the downwind plume (as illustrated in Figure 3). The Bayesian inversion outperforms the least-squares method in estimating $NO_x$ emissions with more accurate and consistent results while avoiding the bimodal errors of the least-squares approach. Lifetimes, however, remain more challenging to reproduce and show mixed results between the two methods. The sensitivity tests show that also with deviating prior information, the Bayesian inversion method outperforms the Least-Squares approach, producing smaller biases and a smaller standard deviation.

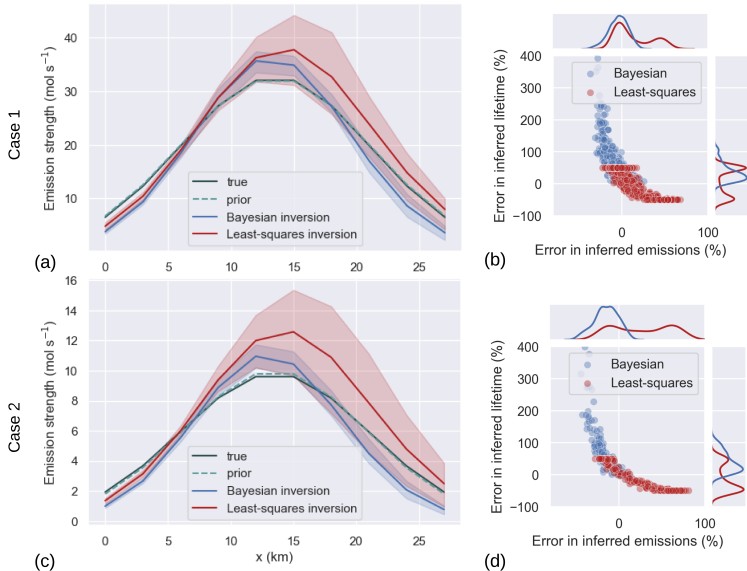

**Figure 4.** a,c) Mean emission strength of each 5 km line cell along the city for an ensemble of 1000 randomly generated noise profiles over the city for both cases. The shaded areas indicate the interquartile range. b,d) The correlation between the deviation from the true $NO_x$ emissions and lifetimes, with kde (kernel density estimate) plots in the margins. For the Bayesian inversion (blue) and the least-squares inversion (red).

## 4   Model application: $NO_x$ emission estimations Paris

We now apply the two inversion methods of the superposition column model (the least-squares method and the Bayesian method) to daily clear-sky TROPOMI $NO_2$ data to estimate $NO_x$ emissions of Paris between June 2018 - July 2023 at the TROPOMI overpass time of around 13:00 local time. We use the European TROPOMI $NO_2$ product that uses CAMS a priori $NO_2$ profiles in the air mass factor and averaging kernel calculation. This product is based on the operational TROPOMI $NO_2$ (v2.4.0) version, and is described in Douros et al. (2022). We do not apply averaging kernels (Eskes and Boersma, 2003), because the superposition column model does not provide tropospheric $NO_2$ profiles. The standard and European $NO_2$ product has been compared with ground-based remote sensing measurements of nine Multi-AXis Differential Optical Absorption Spectroscopy (MAX-DOAS) instruments by Douros et al. (2022). They found an average bias over all stations of the standard TROPOMI version of -31%. For the European product, this bias is -19%. As compared to the standard S5P tropospheric $NO_2$ column data, the overall bias of the European product for almost all stations is 5 % to 18% smaller, with $NO_2$ columns up to 30% higher than previous in emission hotspots, especially in Winter (Douros et al., 2022). This provides good confidence in using TROPOMI tropospheric $NO_2$ columns from the European product for the purpose of estimating $NO_x$ emissions.

A quality check is applied where less than 10% of the TROPOMI pixels in the study area are allowed to be below a quality assurance (QA) value of 0.75 (recommended by van Geffen et al. (2022)). In this way, cloudy days or problematic retrievals

are filtered out.

## 4.1 Inferring NO$_x$ emissions from TROPOMI NO$_2$ columns

### 4.1.1 Computation of TROPOMI line densities

For the calculation of the line densities, the TROPOMI NO$_2$ data is first rotated towards the effective wind direction (elaborated
in the next section) and re-scaled into grid cells of 0.05×0.05°. Specifically, we do this by generating a target grid with a 0.05°
× 0.05° resolution, aligned parallel to the wind direction at the time of the TROPOMI overpass. The TROPOMI NO$_2$ data are
then regridded onto this new grid, using weights based on the overlapping areas between the original and target grids.

Within the 65×65 km study domain, the grid cells are divided into 13 'line cells' along the wind direction, as illustrated
in Fig. 1 for the TROPOMI overpass on 18 May 2018. Subsequently, the line densities are calculated by accumulating all the
285 pixels within each line cell, and dividing by the total width. The result is one value of the NO$_2$ line density for each 'line cell',
with units of mol/cm. This is a transformation of a 2-D (65×65 km$^2$) field into a 1-D line density, which simplifies the analysis,
at the cost of giving up any constraints on the across-wind emission distribution.

The result of this line density transformation is shown for the overpass on 18 May 2018 in the right panel of Fig. 1. Here,
the line densities are shown for each distance, where $x = 0$ is the upwind start of the area. In the case of this example, the line
densities are increasing until 40 km, followed by a slight decay. This pattern arises from higher emissions in the center of Paris,
at $x{\sim}30$ km, after which emissions are lower and decay of NO$_2$ dominates (Lorente et al., 2019).

### 4.1.2 Estimating NO$_x$ emissions and lifetimes

To compute the NO$_x$ lifetimes and emissions across the city, the superpositon model (Eq. 1) is fitted to the calculated TROPOMI
NO$_2$ line densities using the two different inversion methods: the least-squares method and the Bayesian variant.
The background NO$_2$ level ($b$) is defined as the average line density value in a box of 30 km upwind of the study area (light
grey dotted box in Fig. 1). In this definition, chemical loss of background NO$_2$ by reaction with OH is neglected, because the
background NO$_2$ is assumed to be mostly located above the boundary layer where OH concentrations are assumed to be lower
than in the photochemically active boundary layer. The domain average, boundary layer mean NO$_x$/NO$_2$ ratio over Paris is taken
from CAMS (0.4°×0.4°resolution), 1h before the TROPOMI overpass time. The CAMS 0.4°forecast product is part of the
300 Copernicus Atmosphere Monitoring Service and provides global 5-day forecasts of atmospheric composition at approximately
0.4°( 40 km) resolution. Similarly, we take CAMS boundary layer mean OH as prior in our Bayesian inversion. Using CAMS
OH as a prior is justified by its physical consistency, full spatiotemporal coverage, and compatibility with the scale of the
rotated line densities and column model, which assumes a single effective NO$_x$ lifetime. The domain and boundary layer
average wind speed, weighted by the vertical NO$_2$ concentration from CAMS, is taken for 1h before the TROPOMI overpass
from ERA5, the fifth-generation ECMWF (European Centre for MediumRange Weather Forecasts) atmospheric reanalysis of
the global climate.

### 4.1.3 Prior estimates and uncertainties

The TNO-MACC-III $NO_x$ emission inventory of 2011 (Kuenen et al., 2014) is used as a prior estimate of the $NO_x$ emissions over Paris. This inventory predicts a total prior $NO_x$ emission of 52.8 mol s$^{-1}$ over Paris for 2011. We scale this value of 2011 to the years 2018-2023 using predicted $NO_x$ emission reductions after 2011 of France by the EEA ranging from -27% for 2018 to -49% for 2023 (EEA, 2023). In 2020 and 2021, France took measures to prevent the spread of the coronavirus outbreak (Covid-19), which caused reductions in industrial activities and traffic intensity. To correct for this decrease in activity, we account for an additional decrease in the prior emissions of 40% during the three Covid-19 lockdown periods of France (Guevara et al., 2021; Ikhlasse et al., 2021). The $NO_2$ concentrations in Paris never completely increased to their pre-Covid levels in between the lockdown periods (Pazmiño et al., 2021), which is why we assume that prior emissions were reduced by 20% in between the lockdown periods. A timeseries of these prior $NO_x$ emissions is displayed in the light green line of the upper panel of Fig. 5.

For a more realistic prior value than one yearly average, we scale the emissions using monthly, weekly and hourly emission factors from TNO (Denier van der Gon et al., 2011), based on prior knowledge of human activities. These factors are given per source sector. We weigh the temporal emission factors by the contribution of the source sectors of Paris according to the European Union (EU, 2019). This gives us an hourly scaling factor of 1.17 for 12:00 (around the TROPOMI overpass time), because of higher traffic intensity at 12:00 relative to the 24-hour mean values. We also apply additional scaling factors varying per month and weekday. These are higher for Winter months, when there is more residential heating, and vehicles suffer more from so-called cold starts (Tu et al., 2021) and lower for weekend days because of reduced traffic intensity. A timeseries of this corrected prior $NO_x$ emission, including the scaling factors, is displayed in the dark green line of the upper panel of Fig. 5. It should be noted that our constructed prior is intentionally simple and straightforward. The prior only needs to be a good starting point, but the observations will adjust the posterior emissions towards the most accurate solution.

For the prior estimate of the $NO_x$ lifetime, the domain-average, boundary layer mean hydroxyl radical (OH) concentration is taken from CAMS for one hour before the TROPOMI overpass.

In the Bayesian inversion method, $NO_x$ emissions and lifetimes are permitted to deviate from the prior, constrained by observation and prior uncertainties. We use a line density observation uncertainty $\sigma_o$ of 10%, accounting for both the measurement uncertainty and the uncertainty of the model representation of the system. To incorporate the uncertainty in OH concentrations and its impact on the $NO_x$ lifetimes, we choose a standard deviation of 30% on the prior lifetime ($\sigma_{A,\tau}$). This selection aligns with the typical range of uncertainty observed in $NO_x$ lifetimes, which commonly falls within 50% (Lorente et al., 2019). By adopting a standard deviation of 30%, we encompass the majority of uncertainties within the expected 50% range, while also allowing for larger deviations in exceptional cases. Finally, to account for uncertainty in the TNO-MACC-III inventory, we assign a standard deviation of the prior emissions ($\sigma_{A,E}$) of 30% in each individual point (Kuenen et al., 2014)).

## 4.2 NO$_x$ emission estimations Paris

### 4.2.1 Emission estimates 2018-2023

For the period spanning May 2018 to July 2023, we obtained 752 TROPOMI NO$_2$ retrievals over Paris that are under predominantly clear-sky conditions and of sufficient quality to perform NO$_x$ emission estimates. Some of these estimates are in duplicate because Paris is observed from two subsequent satellite overpasses on some days. This results in 560 inversions corresponding to unique days. Initially, the emission inversions were conducted with both the Bayesian method and the method outlined by Lorente et al. (2019). The time-averaged estimates of both methods were similar, but the latter revealed substantial outliers in NO$_x$ emission estimates, reaching up to 250 mol s$^{-1}$ on some days, and lifetimes were underestimated, particularly on days with low wind speed. This underscores the overfitting issue that we raised in the previous section. We therefore continue analysing the NO$_x$ emission estimates from the Bayesian inversion. Table 4 shows the average conditions across all inversions.

**Table 4.** Average meteorological and chemical conditions over Paris throughout the period May 2018 - July 2023, and prior and posterior NO$_x$ emissions and lifetimes during the TROPOMI overpass time (around 12:00). NO$_x$/NO$_2$ ratios and wind speeds are derived from CAMS, and temperatures are measurements from the Montsouris weather station in Paris. Averages are given for the whole period and for the Summer and Winter months separately, the range is one standard deviation. The posterior standard deviation is estimated through a Monte Carlo analysis, using 50 randomly drawn prior and observation values, with their prescribed uncertainties as standard deviations, for eight distributed days in 2022.

| | n | NO$_x$/NO$_2$ | Wind speed (m s$^{-1}$) | Temperature (°C) | NO$_x$ lifetime (h) | | NO$_x$ emissions (mol s$^{-1}$) | |
|---|---|---|---|---|---|---|---|---|
| | | | | | Prior | Posterior | Prior | Posterior |
| Year-round | 752 | 1.40 ± 0.12 | 4.6 ± 2.3 | 20.4 ± 8.1 | 5.2 ± 30% | 6.0 ± 3.9% | 35 ± 30% | 32 ± 6.1% |
| Winter (DJF) | 114 | 1.53 ± 0.19 | 5.4 ± 2.8 | 9.5 ± 4.4 | 30 | 27 | 39 | 40 |
| Summer (JJA) | 243 | 1.33 ± 0.09 | 4.0 ± 1.9 | 28.4 ± 3.8 | 2.7 | 2.3 | 31 | 28 |

The CAMS-derived domain average NO$_x$/NO$_2$ ratios, averaging to 1.4 over all inversions, exceeded the commonly adopted ratio of 1.32. This discrepancy results in higher NO$_x$ emission estimates than if the constant value of 1.32 would be used. Daily temperatures, recorded at 13:00 from the Montsouris weather station in Paris city center, consistently appear relatively high, potentially influenced by urban heat island effects and our clear-sky sampling. Our findings indicate slightly lower average NO$_x$ emissions than the prior estimates, especially during the Summer months. We find an average top-down NO$_x$ emission over Paris of 32 mol s$^{-1}$, which is slightly lower than the prior (9% year round), especially during the Summer months (11%).

The middle panel (b) of Fig. 5 displays the monthly average NO$_x$ emissions of Paris estimated with the Bayesian inversion method (blue line). Values from the CAMS-REGv7 inventory are displayed in red. This is an improved version of the v4 dataset described by Kuenen et al. (2022). We scale these values using emission factors from Guevara et al. (2020, 2021) (more information about how we calculated these values can be found in section 3 of the Supplementary Material). To ensure a fair comparison, the prior estimate in this graph is resampled on days with valid inversions, resulting in a slight variation from the upper panel. The monthly average posterior NO$_x$ emissions exhibit more variability than the prior. Higher variability

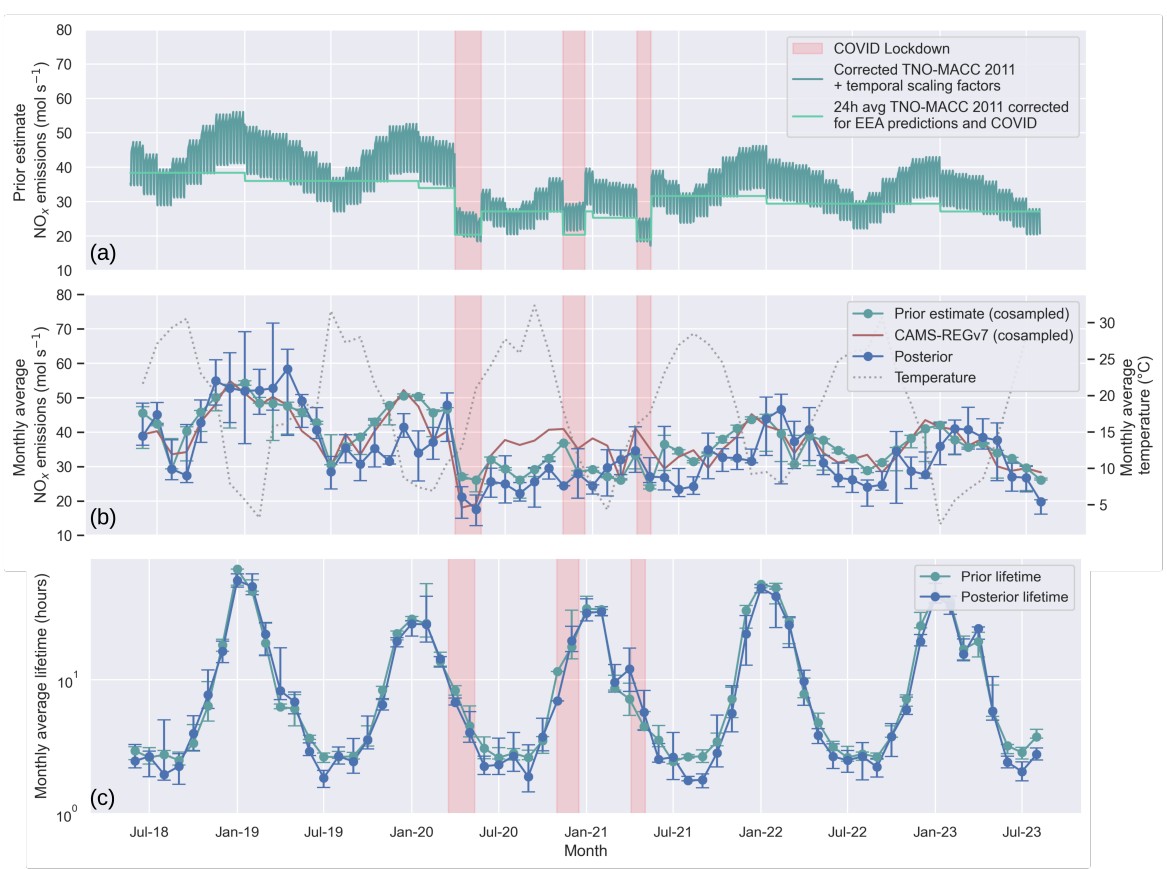

**Figure 5.** Upper panel: NO$_x$ emissions from the TNO-MACC III emission inventory, corrected for the emission reductions predicted by the EEA, and for emission reductions during COVID (light green). The dark green line shows daily predictions, accounting for the weekly and monthly cycle and uses a scaling factor of 1.17 (relative to the 24-hour mean) for the TROPOMI overpass time. Lower two panels: monthly median values of the prior (green) and posterior (blue) NO$_x$ emission (b) and lifetime (c) estimates. The prior is resampled to the days with TROPOMI NO$_2$ retrievals. The errorbars represent the interquartile range within each month.

of posterior emissions is expected because of uncertainties in their derivation. Additionally, posterior emissions reflect real day-to-day and diurnal fluctuations, while prior emissions are based on climatological averages and are therefore inherently less variable. This difference between prior and posterior NO$_x$ emissions indicates that factors beyond the month and day of the week influence the emissions.

We observe an overall decreasing trend from 44 to 32 mol s$^{-1}$ (27%) in NO$_x$ emissions between May 2018/19 and August 2022/23. This decreasing trend can be partly attributed to the Paris low-emission zone, which was estimated to reduce traffic NO$_x$ emissions by about 20% between 2018-2023 due to the adoption of cleaner vehicles (Bernard et al., 2020). Significantly lower NO$_x$ emissions are visible during the Covid-19 lockdown periods, even lower than our prior assumptions. Especially

during the first Covid-19 lockdown (17/03/2020-11/05/2020) the $NO_x$ emissions dropped substantially. We quantified the effect of the Covid-19 lockdowns by calculating the change in emissions between the lockdown periods in 2020-2021 and the prior of the same periods in 2019. We find a significant decrease in the posterior for the first lockdown to 17.5 mol s$^{-1}$, a reduction of 61%, surpassing the prior prediction of 44%. The second lockdown exhibits a reduction of around 40% compared to the 2019 prior for both the prior and posterior estimates. In the last lockdown, the decrease is less intense at 37% compared to the prior's 49%.

The lower panel of Fig. 5 shows the time series of prior and posterior $NO_x$ lifetime estimates over Paris. Our Bayesian inversion framework captures seasonality, but the retrieved lifetime values should not be overinterpreted as chemically precise quantities. Our end-to-end test (Table 3 and Fig. 4) showed that the lifetime retrievals are subject to significant biases—up to 30%—highlighting the limitations of the method. This bias stems from the inherent asymmetry in the inversion sensitivity: the $NO_2$ line density is strongly and directly influenced by the strength of the $NO_x$ emissions, whereas the lifetime exerts a more subtle control through the dampening of the increase in line densities with distance. In practice, the signal from $NO_x$ emissions dominates the inversion, while the $NO_x$ lifetime estimate is more a regularization parameter that prevents overfitting than a robust diagnostic of possible changes in atmospheric chemistry.

Figure 5 shows good agreement between prior and posterior lifetime estimates for most months, but in summertime the posterior lifetimes are often significantly below the prior values. This indicates that the $NO_x$ emission reductions from 2018 to 2023 are not only supported by the satellite-observed changes in $NO_2$ column densities, but also require shorter effective lifetimes in the inversion to fit the observed spatial gradients. Taken together with the fact that posterior $NO_x$ emissions are consistently lower than prior values, this points unambiguously to a substantial reduction in $NO_x$ pollution over Paris. The simultaneous decrease in both posterior emissions and lifetimes, relative to the prior, reinforces the robustness of this conclusion: the observed $NO_2$ pattern cannot be reconciled without assuming cleaner conditions than those represented by the prior.

### 4.2.2 Seasonal and weekly cycle

In Winter, enhanced $NO_x$ emissions are expected due to engine cold starts and increased residential heating demand (Paris generates the energy for heating within the city itself). Our analysis reveals a distinct seasonal cycle of $NO_x$ emissions with a Winter:Summer ratio of 1.38 (Fig. 6a). Contrastingly, in the prior estimate, the fall months show the highest emissions (1.28 compared to the Summer). This could suggest that residential heating starts later than expected in the prior. We find that $NO_x$ emissions are generally overestimated in the prior in Summer and underestimated in Winter. This is in line with the study of Lorente et al. (2019), who found that bottom-up emission inventories underestimate actual residential heating emissions in Winter months.

Additionally, a significant difference is found between $NO_x$ emissions during low temperatures (<10°C) and high temperatures (>20°C) (Fig. 6b). We filtered the data by excluding weekends, lockdown periods and the Summer holiday period (July-August) to mitigate potential holiday effects. The resulting average posterior difference between low and high-temperature emissions is 43-35 mol s$^{-1}$, a difference of 23%. This represents a similar but slightly stronger difference compared to the prior (43-33 mol s$^{-1}$).

We observe a distinct weekly cycle in Paris (Fig. 6c), starting with low emissions on Mondays, elevated levels on Thursdays and Fridays, and a reduction during the weekend (25%). This cycle is slightly more pronounced than initially assumed in the prior, which predicts a weekend reduction, calculated relative to the Mon-Fri average, of 22%. This weekend reduction is smaller than what was found by Lange et al. (2022) (40%) and Lorente et al. (2019) (35%). In Summer, the decrease in $NO_x$ emissions in the weekend is much larger (39%) than in Winter (11%). This is likely because of a higher contribution of traffic emissions to the total emissions in the Summer months, whereas in Winter the share of traffic emissions may be smaller because of local residential heating and power generation. In Winter, the posterior weekend reduction is lower than in the prior inventory. This, again, could point to a prior underestimation of residential heating. Additionally, the weaker weekly emission cycle observed in winter could be influenced by the effect of cold starts. On weekdays, vehicles are typically started early in the morning, while on weekends, car usage tends to begin later, closer to the TROPOMI overpass time. Weekend day emissions could then show up higher than without cold starts, dampening the weekly cycle.

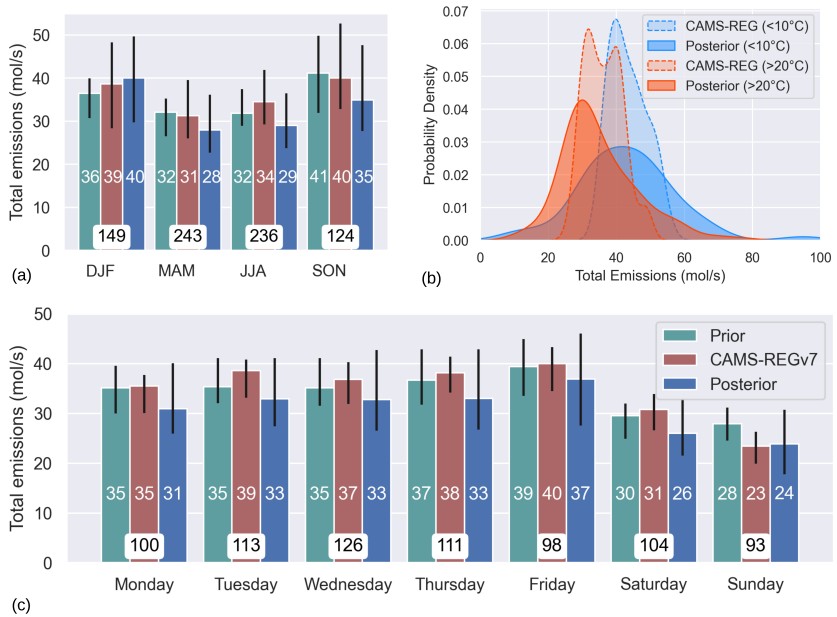

**Figure 6.** The seasonal (a) and weekly cycle (c) of $NO_x$ emissions in Paris, of the prior, the posterior, and the CAMS-REGv7 inventory. White boxes display the count within each category. (c) shows the CAMS-REGv7 and Posterior $NO_x$ emissions, grouped by temperature (<10°C (n=56) and >20°C (n=189). The weekends, Summer holidays and COVID lockdowns are filtered out here.

### 4.2.3 Literature comparison

In Fig. 7, a comparison between prior and posterior emissions is presented alongside multiple inventory datasets. The interannual variation reveals a slight decrease in emissions in 2019, followed by a substantial decline in 2020 during the implementation of Covid-19 restrictions. Emissions veered back in 2021 and stabilized through 2023.

We compared our findings with other literature that estimates $NO_x$ emissions in Paris (Fig. 7). Lorente et al. (2019) reported higher $NO_x$ emissions for 2018, but only investigated these between January and June, while our estimation started from May 2018 onwards. We incorporated a corrected reaction rate constant for the oxidation of $NO_2$, which could contribute to the divergence in estimates. Lange et al. (2022) also estimated $NO_x$ emissions for Paris from 2018 to 2020, reporting an average emission of 56.2 mol s$^{-1}$. Discrepancies here may arise from a different estimation method or variations in the definition of the Paris city area. Our emission levels align more closely with those reported by Lonsdale and Sun (2023). Their findings, presented in nmol m$^{-2}$ s$^{-1}$, were converted to match our surface area unit. They observed lower values for 2019, comparable to our study for 2020 and 2021, and slightly lower values for 2022. The use of a fixed $NO_x/NO_2$ ratio of 1.32 could contribute to their slightly lower emissions.

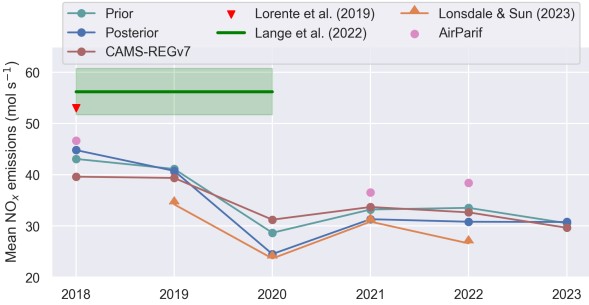

**Figure 7.** Annual $NO_x$ emissions over the Paris from 2018 to 2023. For our study (blue) and other studies. Note that our analysis spans from May 2018 to July 2023, so the averages of 2018 and 2023 are not over the whole year.

We also compare our results to the CAMS-REGv7 inventory and $NO_x$ emission estimates from AirParif, the air quality observatory in the Île-de-France region (AirParif, 2021, 2023). AirParif provides emission estimates averaged across the entire Île-de-France area, which we scaled to align with our Paris study domain. Given that $NO_x$ emissions are higher closer to the city center and lower in outlying areas, we applied a scaling factor of 1.7 based on the ratio of emissions from the CAMS-REGv7 gridded inventory between the full Île-de-France region and our study area. These adjusted annual emission values are presented as pink dots in Fig. 7.

Since these values represent annual averages, they are not co-sampled with TROPOMI overpasses, which could explain the slightly higher $NO_x$ emission estimates from AirParif compared to our estimates (blue line). For instance, TROPOMI overpasses with sufficient cloud-free conditions are generally more frequent in Summer, which biases our annual averages toward this season, when $NO_x$ emissions are typically lower.

# 5 Discussion and conclusion

We evaluated the performance of our superposition column model (Lorente et al., 2019) in estimating $NO_x$ emissions within urban areas using satellite observations. Our investigation analyzed the ability of the forward model to calculate $NO_2$ columns over cities, given its use of simplified temporally and spatially averaged $NO_x$ emissions, wind speed, $NO_x$ lifetime, and $NO_x/NO_2$ ratios. We performed a comparison to synthetic $NO_2$ observations generated with the high-resolution Large Eddy Simulation model MicroHH, which simulates atmospheric dynamics and chemistry over a hypothetical city of 30 km $\times$ 30 km in minute detail. MicroHH simulates substantial variability in $NO_x$ lifetime (1-9 h) and $NO_x/NO_2$ ratio (1.2-1.7) over the city domain, but the absolute deviation between $NO_2$ line densities simulated with the superposition model and with MicroHH stayed within 7%. This indicates that the superposition model is effective in describing the evolution of column $NO_2$ with distance over a large city, given known average $NO_x/NO_2$ and OH concentrations, despite averaging variable chemical and meteorological parameters over the city domain.

Tests with inferring $NO_x$ emissions from synthetic $NO_2$ line densities simulated by MicroHH using the superposition model showed that simply minimizing the least-squares using a look-up table approach, as was done before in Lorente et al. (2019) frequently resulted in overfitting, where the $NO_x$ lifetime is overestimated and the $NO_x$ emissions are underestimated or vice versa. We propose a more formal Bayesian approach of the inversion of the $NO_x$ emissions, which not only considers the fit to the observations, but also incorporates prior information about $NO_x$ emissions and lifetime, to keep the solution in check. Although the Bayesian approach exhibits slightly larger discrepancies between the modeled and observed line densities, it yields solutions closer to the a priori known MicroHH emissions. The Bayesian approach reproduces the known $NO_x$ emissions to within 4%, whereas the least-squares minimization, which does not take into account uncertainties in the prior emissions, reproduces emissions to within 20%.

We applied Bayesian inversion to infer a 5-year time series of daily $NO_x$ emissions for Paris using TROPOMI $NO_2$ V2.4.0 retrievals from June 2018 to August 2023 under clear skies. Incorporating prior emission estimates from the TNO-MACC-III inventory, corrected for France's emission decrease reported by the European Environmental Agency, we found average $NO_x$ emissions of 32 mol s$^{-1}$, which is 9% lower than the prior estimate. We observe an overall reduction in $NO_x$ emissions between 2018 and 2023 of 27%, compared to a reduction of 12% between 2018 and 2023 in the prior estimate. COVID-19 lockdowns led to sharp reductions of 61%, 40%, and 37% during the first, second, and third lockdown relative to emissions in the same period of the year before the COVID-19 measures. We observed a Winter:Summer emission ratio of 1.38, and significantly higher $NO_x$ emissions on days with lower temperatures in Paris. We find a weekend $NO_x$ reduction of 25%, slightly more pronounced than the weekend effect of 22% in the emission inventory. We demonstrated that the improved Bayesian inversion method of the superposition model offers a reliable and efficient means to monitor daily $NO_x$ emissions and evaluate policies in urban areas.

*Code and data availability.* The TROPOMI L2 product used in this study is available through the TEMIS portal (https://www.temis.nl/airpollution/no2col/no2_euro_tropomi_cams.php, last access: 6 January 2025). CAMS model data were retrieved from the CAMS Atmosphere Data Store (https://ads.atmosphere.copernicus.eu, last access: 6 January 2026) and its predecessor hosted by Météo-France. The MicroHH code used for the calculations is available from GitHub (https://github.com/microhh/microhh, branch main_kpp, last access: 6 January 2025). The inventory from TNO (TNO-MACC-III) is available on request by contacting HDG. Access to the CAMS-REG-v7 is provided through the Emissions of atmospheric Compounds of Ancillary Data (ECCAD) system. Since the ECCAD system requires a registration and login, a sample of the emission files has been made available for download directly. This sample includes data for the year 2017 and is available through https://eccad.aeris-data.fr/essd-surf-emis-cams-reg/ (last access: 6 January 2025). Daily $NO_x$ emission estimates will be made available through zenodo (Mols, 2025).

*Author contributions.* AM and KFB designed the study. AM performed the data analysis with support from KFB. AM wrote the manuscript with contributions from KFB, MK and HDG. MK performed the simulations in MicroHH. HDG provided the bottom-up inventory and guidance on the interpretation thereof

*Competing interests.* The authors declare that they have no conflict of interest.

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
