# Peer review of "An improved Bayesian inversion to estimate daily NOx emissions of Paris from TROPOMI NO2 observations between 2018-2023"

_EGUsphere, 2025_

## Author Comment (AC1)

Authors' response to referee comments #1 regarding 'An improved Bayesian inversion to estimate daily NOx emissions of Paris from TROPOMI NO2 observations between 2018-2023' by Mols et al. (2025).

The reviewer's comments are in black, the authors' replies in blue.

**Reviewer #1**

This article presents an new, improved method for estimating NOx emissions over urban areas based on TROPOMI or other high-resolution spaceborne NO2 data. The method builds on a previous method (Lorente et al. 2019) but introduces a well-thought Bayesian framework for the optimization of NOx emissions and lifetimes. In this way, the various uncertainties are taken into consideration, and overfitting is avoided. The advantages of the method are shown by tests (OSSE) using synthetic observations generated by a high-resolution model. Next, the method is applied to the estimation of NOx emissions over Paris using TROPOMI data. The results lead to several interesting insights on the emissions, including their trends, seasonal and weekly cycles, and variability due to covid-19 lockdowns. Overall, the manuscript is well-written, the methodology is clearly presented, with a few minor reservations (see below), and the results appear robust and useful to the top-down emission community. I see no reason why this method could be applied to many other cities and industrial centers worldwide. I recommend publication in this journal, provided that the authors address the following minor comments listed below.

We thank the reviewer for their encouraging words and suggestions for additions to the manuscript. Please see below for replies to the specific comments.

**Minor comments**

l. 11-12 and l. 321: The decrease is -27% based on the 2018 and 2023 totals, not 17.5 or 18%. Please clarify.

We are glad that this reviewer spotted this error. This value was indeed wrong, we changed it to -27%, also in the results and conclusions.

Abstract and Conclusions: Can this method be applied to other large cities or industrial centers? A bit of discussion would be welcome.

We agree that a discussion about the applicability of the method to other NO$_x$ sources is a good and needed addition to our manuscript. We added the following paragraph at the last part of the conclusion:

*"In the future, the superposition model can be applied to estimate NO$_x$ emissions from other large cities or industrial centers, provided that emissions from a given source are clearly attributable to that source. This requires that the NO$_2$ plume signal exceeds the detection threshold, and that the origin of the NO$_2$ plume can be linked to a spatially distinct city or emission source. In cases where multiple plumes from different sources overlap, the current model is not applicable. However, future model developments may allow for the separation of overlapping emission signals from multiple sources. Also, with new, high resolution geostationary satellites, it will become easier to attribute NO$_2$ plumes to specific sources. For example, ESA's TANGO mission, scheduled for launch in 2027, will detect NO$_2$ at a spatial resolution of 300 × 300 m over Europe,*

*enabling much more detailed information on emission sources and their variability (Landgraf et al., 2020)."*

l. 23 NO2+OH is not the only major sink, also formation of PAN (for example) might be important in VOC-rich areas. PAN and other compounds may play the role of NOx reservoirs, which might partly invalidate the assumptions of the superposition model. I think that this issue should be mentioned and possibly discussed.

The superposition model is designed to estimate effective $NO_x$ emissions and lifetimes from TROPOMI $NO_2$ enhancements over urban areas. It is not intended to explicitly resolve all chemical pathways, but rather to capture the dominant processes controlling $NO_x$ removal on spatial and temporal scales relevant to TROPOMI retrievals.

We note that (except in cold conditions) PAN formation is not a permanent sink for $NO_x$. PAN is a reversible $NO_2$ reservoir: $CH_3O_2 + NO_2 + M <-> PAN + M$ (e. g. Fischer et al., 2014). Once formed, PAN decomposes rapidly in the warm urban air masses and releases $NO_2$ at the timescale of minutes to an hour. Moreover, the city centre of Paris is generally VOC-limited (e.g. Johnson et al., 2024), and the dominant sink for $NO_2$ in these conditions is oxidation to nitric acid.

We agree that the issue deserves to be mentioned, and we do that now in section 2.1 right after introducing the rate constant of daytime chemical $NO_x$ loss: "PAN formation is not explicitly considered in this framework, as it is a reversible $NO_x$ reservoir rather than a permanent sink:(e.g., Fischer et al., 2014). In the warm, VOC-limited conditions typical of central Paris (e.g., Johnson et al., 2024), PAN decomposes rapidly and contributes little to net $NO_x$ loss. The dominant $NO_2$ sink under these conditions is oxidation to $HNO_3$.

l. 24 Dry and wet deposition of HNO3 are about equally important sinks (see e.g. https://doi.org/10.1029/2018JD029133)

We have added dry deposition as well here: *"Due to its high water solubility, $HNO_3$ is efficiently removed from the atmosphere, primarily through precipitation and direct deposition onto surfaces (Seinfeld and Pandis, 2016)."*

l. 90 Why not adopt a temperature-dependent rate for NO2+OH? The rate is higher in cold conditions (~10% higher at 283K compared to 298K)

We calculate the reaction rate of the $NO_2 + OH + M -> HNO_3 + M$ reaction using the rate constants from Burkholder et al., 2020 (page 434). We use the following equation to obtain the second-order rate constant for a certain temperature and pressure. We use a total gas concentration [M] at 1atm=$2.5*10^{19}$ molecules cm$^{-3}$.

$$k_f([M],T) = \left( \frac{k_o(T)[M]}{1 + \frac{k_o(T)[M]}{k_\infty(T)}} \right) 0.6^{\left\{ 1 + \left[ \log_{10}\left( \frac{k_o(T)[M]}{k_\infty(T)} \right) \right]^2 \right\}^{-1}}$$

| | $k_0$ [cm³/molecule/s] | $K_\infty$ [cm³/molecule/s] | K (1atm, T) [cm³/molecule/s] |
|---|---|---|---|

| 270K | $2.74 \times 10^{-30}$ | $2.5 \times 10^{-11}$ | $1.193 \times 10^{-11}$ |
|------|------------------------|-----------------------|--------------------------|
| 288K | $2.26 \times 10^{-30}$ | $2.5 \times 10^{-11}$ | $1.101 \times 10^{-11}$ |
| 298K | $2.04 \times 10^{-30}$ | $2.5 \times 10^{-11}$ | $1.053 \times 10^{-11}$ |

In our model, we use k' = $1.1 \times 10^{-11}$ cm$^3$/molecule/s, so the rate constant for 288K, a quite average yearly daily max temperature for Paris. Indeed, as the reviewer indicates, the rate constant is higher at colder conditions, around 8% higher for 270K, and it is lower for higher temperatures, 4.5% lower for 298K. This temperature dependence on the reaction rate constant is relatively small, and we only use it to calculate a prior estimate of the reaction rate, which we fit later together with the $NO_x$ emissions using our inverse model. We assume that this small error is captured by the 30% uncertainty that we apply during the inversion.

l. 206-208  Based on Fig. 4, the prior is very close to the truth. Why is that? This might contribute to explain why the Bayesian inversion results are closer to the truth, due to the constraint from the first term of the cost function (Eq. 3).  What would happen if the prior was more different from the truth?

The reviewer raises an important point here. Our primary goal with the OSSEs for Symcity was to assess whether the simple inversion method can reliably infer emissions and lifetimes when the prior is accurate. However, we acknowledge that in realistic scenarios, prior information is not known with a high degree of certainty. To investigate the sensitivity of our results to deviations in the prior, we conducted an additional test. Specifically, we repeated the OSSEs for both Symcity cases 50 times, introducing a ±20% deviation in either the prior lifetime or emissions. We then analyzed the performance of both the Bayesian and Least-Squares inversion methods.

In the first sensitivity test, we evaluated the accuracy of the inferred $NO_x$ emissions and lifetimes when the prior emissions were biased by 20%. The resulting posterior emission and lifetime deviations (calculated as $\frac{posterior - true}{true} * 100\%$) for both inversion approaches are presented in Table 1 below. The results from the least-squares inversion are not dependent on the prior emission and thus remain unchanged from the case with a known prior. For the Bayesian inversion, we used the same uncertainty settings as described in Section 3.2 of the manuscript.

|  | Least-squares | Bayesian | | |
|---|---|---|---|---|
|  | Prior = True values | Prior = True values | Prior E +20% | Prior E -20% |
| Emissions case 1 | 14% ± 22% | -1% ± 7% | 0.8% ± 6.3% | -6.7% ± 6.2% |
| Emissions case 2 | 26% ± 34% | -5% ± 11% | -4.6% ± 10% | -11% ± 6.8% |
| Lifetimes case 1 | 2% ± 46% | 32% ± 31 % | -1.6% ± 8.2% | 31% ± 24% |
| Lifetimes case 2 | -19 ± 37% | 13% ± 61% | 2.5% ± 25% | 22% ± 45% |

Table 1: Errors in the posterior $NO_x$ emissions and lifetimes inferred using the 2 inversion methods, using prior emissions that deviate ±20% from the true emissions. Errors are  (calculated as (posterior-true)/true*100%).

We did the same sensitivity test for a bias in the prior $NO_x$ lifetimes. The results for both inversion approaches are shown in Table 2 below.

|  | Least-squares | Bayesian |
|---|---|---|
|  |  |  |

|  | Prior known | Prior tau +20% | Prior tau -20% | Prior known | Prior tau +20% | Prior tau -20% |
|---|---|---|---|---|---|---|
| **Emissions case 1** | 14% ± 22% | 8.4% ± 17% | 13% ± 20% | -1% ± 7% | -4.8 ± 7.6 | 2.1% ± 6.0% |
| **Emissions case 2** | 26% ± 34% | 7.4% ± 26% | 29% ± 35% | -5% ± 11% | -9.6% ± 7.4 | -0.5% ± 8.8% |
| **Lifetimes case 1** | 2% ± 46% | 4.5% ± 54% | -9.7 ± 34% | 32% ± 31 % | 28% ± 27% | -4.3% ± 18% |
| **Lifetimes case 2** | -19% ± 37% | 2.9% ± 48% | -29% ± 30% | 13% ± 61% | 17% ± 14% | -7.5% ± 17% |

*Table 2: Errors in the posterior $NO_x$ emissions and lifetimes inferred using the 2 inversion methods, using prior lifetimes that deviate ±20% from the true lifetimes. Errors are (calculated as (posterior-true)/true*100%).*

These sensitivity tests show that increasing or decreasing the prior emissions by 20% results in a posterior bias of no more than 6% compared to the case with a known prior. This confirms that the Bayesian inversion method uses both the prior and the observations effectively. Even with deviating prior emissions, the Bayesian inversion method still outperforms the Least-Squares approach, producing smaller biases and a smaller standard deviation. Also when the prior lifetime is varied (Table 2), the Bayesian inversion retrieves posterior emissions much closer to the true values than the Least-Squares inversion. In practice, when applying the superposition model to real cases, one would typically also have some estimate of the prior uncertainty. These uncertainties can be reflected in the values of $\sigma_{a,E}$ and $\sigma_{a,k}$ to prevent the Bayesian inversion from relying too heavily on a potentially inaccurate prior.

Unlike the Least-Squares approach, which fits the line densities directly, the Bayesian method balances observational data with prior knowledge. Even if the prior is not perfectly accurate, it can still help guide the solution in the right direction, leading to more consistent and reliable estimates.

We appreciate that the reviewer raised this point, as it demonstrates the robustness of the Bayesian approach under more realistic conditions. We added this analysis to the supplementary material, and now refer to it in the manuscript at the end of section 3.2: *"To investigate the sensitivity of our results to deviations in the prior, we conducted an additional test. We repeated the OSSEs for both Symcity cases 50 times, introducing a ±20% deviation in either the prior lifetime or emissions. The results can be found in section 2 of the Supplementary Material. These sensitivity tests show that increasing or decreasing the prior emissions by 20% results in a posterior bias of no more than 6\% compared to the case with a known prior."* … *"The sensitivity tests show that also with deviating prior information, the Bayesian inversion method outperforms the Least-Squares approach, producing smaller biases and a smaller standard deviation."*

l. 247 Some more explanation (or maybe a reference) might be needed regarding the rotation and re-scaling step.

We agree, and added some more explanation on this step. We now added the following section on this to the manuscript:

*"For the calculation of the line densities, the TROPOMI NO2 data is first rotated towards the effective wind direction (elaborated in the next section) and re-scaled into grid cells of 0.05x0.05°. Specifically, we do this by generating a target grid with a 0.05° × 0.05° resolution, aligned parallel to the wind direction at the time of the TROPOMI overpass. The TROPOMI NO2 data are then regridded onto this new grid, using weights based on the overlapping areas between the original and target grids."*

l. 264 CAMS NOx data are used for the domain average NOx/NO2 ratio. At what altitude above ground?

We use the boundary layer mean NO and NO2 values for this. We added this to the manuscript.

l. 275 "The NO2 concentrations (...) never completely decreased to the original levels": I do not follow here. Do you mean "increased"?

The reviewer is correct, we changed this. Also, for more clarity we changed '*original*' to '*pre-Covid*' here.

l. 288 What altitude for CAMS OH? Or is it an average weighted by the NO2 profile?

We use the boundary layer mean OH values for this. We added this to the manuscript.

l. 319-320 The higher variability of posterior emissions is expected due to uncertainties in their derivation.

We agree with this, but argue that the higher variability is expected because of 1) uncertainties (as the reviewer points out), but also 2) because posterior emissions reflect real day-to-day and even diurnal variability, whereas prior is inherently less variable because it represents climatological emissions. We therefore added the following lines to the manuscript:

*"The monthly average posterior $NO_x$ emissions exhibit more variability than the prior. Higher variability of posterior emissions is expected because of uncertainties in their derivation. Additionally, posterior emissions reflect real day-to-day and diurnal fluctuations, while prior emissions are based on climatological averages and are therefore inherently less variable. This difference between prior and posterior $NO_x$ emissions indicates that factors beyond the month and day of the week influence the emissions."*

l. 345 and elsewhere in this paragraph: are the weekend reduction calculated relative to the weekly (7-day) average, or relative to Mon-Fri average?

We agree that this is not completely clear in the text and thank the reviewer for pointing this out. The weekend reduction is calculated as the weekend average relative to the Mon-Fri average. We added this clarification to this line.

l. 350-351 I don't see how the higher cold start emissions in winter would reduce the weekend effect. It would be the other way around since traffic emissions are (expected to be) more strongly reduced during weekends. Therefore, only residential heating would have to explain the much weaker weekly cycle in winter compared to summer. Is this reasonable? What are the relative shares of the different sectors in the Paris area?

We thank the reviewer for this comment, and we agree that this section indeed calls for some further discussion. The statement about the colds starts that we give now does not explain the reduced weekend effect in winter. And indeed, in the Paris area, traffic has a share of ~50%, and residential heating ~15% (AirParif, 2021: https://www.airparif.fr/surveiller-la-pollution/les-emissions). This is a year-round average, so the traffic share and residential heating share are closer together in winter, but still residential heating alone can probably not explain the much weaker weekly cycle in winter compared to summer.

We looked further into the cold starts and argue that cold starts in winter dampen the weekend effect because the diurnal cycle of emissions is different on weekend days than on weekdays (see Figure 1 below of the CAMS-TEMPO scaling factors from Guevara et al., 2021). On weekdays, people start their car in the early morning, whereas on weekend days the cars are started on average later in the morning, closer to the TROPOMI overpass time, and therefore this shows up as apparently higher weekend day emissions than otherwise.

[Figure]

*Figure 1: CAMS tempo diurnal scaling factors for weekdays and weekend days*

We added this paragraph to the manuscript:

"In Summer, the decrease in $NO_x$ emissions in the weekend is much larger 39% than in Winter 11%. This is likely because of a higher contribution of traffic emissions to the total emissions in the Summer months. In Winter the share of traffic emissions is smaller because of local residential heating and power generation. In Winter, our posterior weekend reduction is lower than in the prior inventory. This, again, points to an underestimation of residential heating emissions in the prior inventory. Additionally, the weaker weekly emission cycle observed in winter could be influenced by the effect of vehicle cold starts. On weekdays, vehicles are typically started early in the morning, while on weekends, car usage tends to begin later, closer to the TROPOMI overpass time. Weekend day emissions could then show up higher than without cold starts, dampening the weekly cycle."

**Technical / language comments**

l. 6  MicroHH: what does the name stands for? MicroHH is the name of the CDF model itself and is to our knowledge not an abbreviation. But for clarity in the abstract, we added that it is a computational fluid dynamics model.

l. 49 "to estimate the NOx and predict CO2 emissions...": not clear why one is estimated and the other predicted. You could replace by "estimate NOx and CO2 emissions".

This has been corrected as suggested.

Legend of Fig. 1: why "grey arrow"? There are several (apparently) black arrows.

This has been changed to "black arrows".

l. 95 Delete second "on"

 This has been corrected.

Fig. 3 Use same distance units (preferably km) for all panels

The axis units have been changed to km for all panels of Figure 3, as well as Figure 2. l. 139 "the observed NO2 columns"

This has been corrected as suggested.

l. 152 Make a new sentence "It amounts to..."

This has been corrected as suggested.

l. 210 Figure 4b,d (not 4c,d)

 This has been corrected.

l. 243 "Computation of..."

This has been corrected

l. 244 Remove the first sentence since this step is elaborated in the following paragraph.

This has been corrected. We removed this sentence and moved the information about the quality filtering to the previous section.

l. 275 "in between"

This has been corrected.

l. 317 Missing dot after parenthesis.

This has been corrected.

l. 340 Did you really filter data for weekdays? Isn't it for weekends?

We agree that this was phrased unclearly. We changed the phrasing to "We filtered the data by excluding weekends, lockdown periods and the Summer holiday period".

**References author reply**

Burkholder, J. B., Sander, S. P., Abbatt, J. P. D., Barker, J. R., Cappa, C., Crounse, J. D., ... & Wine, P. H. (2020). Chemical kinetics and photochemical data for use in atmospheric studies; evaluation number 19. Pasadena, CA: Jet Propulsion Laboratory, National Aeronautics and Space Administration, 2020.

Fischer, E. V., Jacob, D. J., Yantosca, R. M., Sulprizio, M. P., Millet, D. B., Mao, J., ... & Pandey Deolal, S. (2014). Atmospheric peroxyacetyl nitrate (PAN): a global budget and source attribution. *Atmospheric Chemistry and Physics*, *14*(5), 2679-2698.

Guevara, M., Jorba, O., Tena, C., Denier van der Gon, H., Kuenen, J., Elguindi, N., ... & Pérez García-Pando, C. (2021). Copernicus Atmosphere Monitoring Service TEMPOral profiles (CAMS-TEMPO): global and European emission temporal profile maps for atmospheric chemistry modelling. *Earth System Science Data*, *13*(2), 367-404.

Johnson, M. S., Philip, S., Meech, S., Kumar, R., Sorek-Hamer, M., Shiga, Y. P., and Jung, J.: Insights into the long-term (2005–2021) spatiotemporal evolution of summer ozone production sensitivity in the Northern Hemisphere derived with the Ozone Monitoring Instrument (OMI), Atmos. Chem. Phys., 24, 10363–10384, https://doi.org/10.5194/acp-24-10363-2024, 2024.

Landgraf, J., Rusli, S., Cooney, R., Veefkind, P., Vemmix, T., de Groot, Z., ... & Sierk, B. (2020, May). The TANGO mission: A satellite tandem to measure major sources of anthropogenic greenhouse gas emissions. In *EGU General Assembly Conference Abstracts* (p. 19643).

Seinfeld, J. H., & Pandis, S. N. (2016). *Atmospheric chemistry and physics: from air pollution to climate change*. John Wiley & Sons.

---

## Author Comment (AC2)

The reviewer's comments are in black, the authors' replies in blue

**Reviewer #2**

Title: An improved Bayesian inversion to estimate daily $NO_x$ emissions of Paris from TROPOMI $NO_2$ observations between 2018-2023

Author(s): Alba Mols et al.

MS No.: egusphere-2025-49

**General Comments**

Mols et al. introduce a Bayesian inversion method which determines urban $NO_x$ emissions at daily scale from along-wind line densities. These line densities are produced by integrating TROPOMI $NO_2$ vertical column densities in the cross-wind direction. The study first shows that a simple forward model can represent the relationship between emissions at each cell and the retrieved line densities. Then, a Bayesian approach is introduced where the inversion of this forward model with measured line densities is used to find emissions. Generally, the spatial distribution of $NO_2$ depends on both lifetime and emissions. A significant advantage of this study's approach is the incorporation of prior information on lifetime and emissions into the cost function of the inversion. These priors avoid the overestimation of emissions due to unrealistic representations of the lifetime. The above method is shown to prevent the overfitting of a simpler least-squares inversion, which overpredicted emissions compared to simulated data. The determination of $NO_x$ emissions over Paris between 2018-2023 illustrates interesting effects due to the COVID-19 lockdowns, the low-emission zone, and temperature. The differences between the findings of Lorente et al. and this study are discussed well. The idea is interesting. I recommend publication after attention to the items below.

We thank the reviewer for their insightful comments and suggestions. The points raised have contributed a lot to improving the clarity and quality of the manuscript. Please see below for replies to the specific comments.

**Major comments**

- The assumption that the TROPOMI retrieval is accurate enough to support the authors' analysis should be further explored. The role of a number of resolution dependent aspects of the a priori used in retrievals that would result in systematic biases between city centers and their surroundings have been reported in the literature. It is important to note that these biases always reduce gradients

between peaks in urban plumes and their surroundings. They are not simple random uncertainties. Examples are listed in the references below.

Using low-resolution input in AMF calculations can lead to "resolution dampening," particularly when surface albedo (~50 km) and a priori $NO_2$ profiles (~100 km) are much coarser than the TROPOMI $NO_2$ pixel (~5 km). However, here we use TROPOMI v2.4, which incorporates high-resolution (0.125°) surface albedo from the DLER database (Tilstra et al., 2023) and high-resolution (0.1°) a priori $NO_2$ profiles from CAMS (Douros et al., 2021). These improvements mitigate concerns about insufficient spatial detail. For example, Lange et al. (2023) demonstrated that the v2.3 retrieval using CAMS 0.1° profiles (IUP 2.02.02.01 REG) showed strong agreement with AirMAP $NO_2$ columns in the Ruhr Area. Similar performance is expected over Paris. A corresponding clarification has been added at the end of the introduction.

Nevertheless, errors in the $NO_2$ retrieval are indeed not exclusively random. We recently investigated the issue in Rijsdijk et al. (2025) and found that there likely is a modest degree of spatial error correlation stemming from the surface albedo climatology extending over at least 2 TROPOMI pixels. We have accounted for this in Eq. (6), where we introduce an uncertainty on top of the MicroHH-simulated $NO_2$ columns. This uncertainty that we assign has a random part (originating from measurement noise) and a systematic part (accounting from AMF uncertainties). The systematic part is correlated between adjacent cells, with a Gaussian-like shape between adjacent cells with a spatial correlation length (where the correlation falls to 1/e) of 7km. We added this last clarification to the manuscript in the description of Eq. 6.

- There are many variations of the fitting approach described by Lorente et al in the literature that also aim to reduce the same biases in lifetime and emissions this paper aims to reduce. The paper should include a more complete summary of these approaches and their strengths and weaknesses relative to the stated goals. Recent papers from De Foy, et al. Liu et al, and Zhu et al. are examples, but there are many others.

We have expanded the discussion in the introduction to include a broader range of recent literature on satellite-based estimation of NOx emissions and lifetimes. However, we emphasize that the purpose of this study is not to provide a comprehensive review of existing methods, but to demonstrate and evaluate a specific, observation-driven approach. A full methodological comparison is beyond the scope of this paper.

The introduction was extended as follows: "*These methods have nonetheless been evaluated using synthetic data, with studies such as De Foy et al. (2014) and Liu et al. (2022) showing that inferred NOx emissions and lifetimes remain broadly consistent with the known model input. In a complementary approach, Zhu et al. (2022) inferred long-term changes in NOx lifetime from decadal OMI NO2 observations, using machine learning to relate NO2 columns to OH concentrations.*"

- The paper rightly identifies correlation between $NO_2$ concentration (and emissions) and lifetime as key. It should report on trends in the lifetime with reductions in $NO_2$. These are likely of the same magnitude as the emission reductions but are nonlinear as shown by Zhu et al. (and others). Also, the paper indicates increases in $O_3$ as an important effect on lifetime. The authors should

compare the effect of increased ozone to the effect of differences in $NO_2$ and VOC at the two comparison points. It is likely that an increased source of OH from $O_3$ photolysis is the smallest contributor of these effects, that VOC changes are also small and that $NO_2$ changes dominate.

We appreciate the reviewer's insightful comment regarding possible lifetime trends in the context of changing $NO_x$ emissions. Our Bayesian inversion framework jointly retrieves $NO_x$ emissions and an effective $NO_x$ lifetime by fitting modeled $NO_2$ line densities to satellite-observed line densities. While this setup captures broad trends, we caution against overinterpreting the retrieved lifetime values as chemically precise quantities.

As demonstrated in our end-to-end test (Table 3 and Figure 4), the lifetime retrievals are subject to significant biases --up to 30%-- highlighting the limitations of the method. This bias stems from the inherent asymmetry in the inversion sensitivity: the $NO_2$ line density is strongly and directly influenced by the strength of the $NO_x$ emissions, whereas the lifetime exerts a more subtle control through the dampening of the increase in line densities with distance. In practice, the signal from $NO_x$ emissions dominates the inversion, while the $NO_x$ lifetime estimate is more a regularization parameter that prevents overfitting than a robust diagnostic of possible changes in atmospheric chemistry.

As suggested by the reviewer, we examined trends in both prior and posterior $NO_x$ lifetime estimates over Paris. The CAMS prior suggests a modest increase in lifetime from 2.9 hours in summer 2018 to 3.5 hours in summer 2023. In contrast, our posterior (top-down) estimates indicate relatively stable $NO_x$ lifetimes of 2.7 hours for both summers.

This apparent stability, despite a substantial decline in $NO_x$ emissions, is consistent with the hypothesis of a transition out of the $NO_x$-saturated regime, as proposed by Zhu et al. (2022) and Johnson et al. (2024). In such a regime, $NO_x$ reductions can lead to stable or higher OH levels due to reduced titration.

However, AirParif measurements at the Eiffel Tower (~300 m altitude) show a ~20% decrease in $O_3$ concentrations and no significant change in $NO_x$:$NO_2$ ratios between the Summers of 2018 and 2023. This suggests that while $NO_x$ reductions would favor a shorter lifetime, concurrent decreases in $O_3$ could partly counteract this by limiting OH production. The net effect on OH --and thus on $NO_x$ lifetime-- is likely small and falls within the uncertainty bounds of our inversion framework. We now include a sub-panel in Figure 5 (shown below) showing the time series of prior and posterior $NO_x$ lifetime estimates over Paris to illustrate this point and discuss the implications along the lines of the above text at the end of section 4.2.1.

[Figure]

- The reported improved performance is based on using the domain average lifetime of the simulation as the prior for the inversion. When applying to measured TROPOMI data, the prior lifetime is estimated from the average OH concentration in CAMS. However, a domain average lifetime is a poor approximation to a lifetime that is an explicit non-linear function of NO2.

We acknowledge that using a domain-average OH concentration is a simplification since the effective $NO_x$ lifetime exhibits spatial variability across an urban domain (Figure 3). However, this simplification is explicitly recognized in our framework: the CAMS-based prior is treated as an initial estimate, and we assign substantial uncertainty to it for exactly this reason. The role of the prior is to regularize the inversion, not to dictate its outcome. The posterior lifetime is ultimately constrained by the satellite $NO_2$ observations and reflects the actual spatial $NO_2$ distribution more realistically.

To better support this, we now include in Table 1 the mean and 1-σ spread in $NO_x$ lifetimes from the high-resolution MicroHH simulations, which show about 50% spatial variation in lifetime across the domain (see also Figure 3(a) and 3(d)). This spatial spread confirms that while there is local variability, the domain-mean value remains a reasonable approximation within the context of the uncertainty assigned to the prior.

Therefore, the inversion corrects for biases in the prior, and the posterior reflects the lifetimes consistent with the observed $NO_2$ gradients, providing a robust estimate of effective $NO_x$ lifetime over the city.

- Since the prior is shown to have a significant impact on the inversion, the changing controls on lifetime should be discussed in the context of conclusions on $NO_x$ emissions during different seasons and across long-term trends.

We agree that the prior lifetime influences the inversion outcome and that the chemical controls on $NO_x$ lifetime are a factor in interpreting seasonal and long-term emission trends. As mentioned earlier, we have added a panel to Figure 5 showing the time series of both prior and posterior $NO_x$ lifetime estimates over Paris and include a discussion of their relationship.

Our inversion's sensitivity to lifetime is inherently weaker than to emissions due to the asymmetry in how these two parameters affect $NO_2$ line densities. Emissions influence the absolute magnitude of the column, while the lifetime modulates the downwind gradient more subtly. Our discussion emphasizes that the inversion consistently retrieves shorter posterior lifetimes than the CAMS prior in summer months, and lower posterior emissions year-round. The fact that both quantities decrease relative to the prior strengthens the conclusion of a real and substantial decline in $NO_x$ pollution over Paris from 2018 to 2023. Thus, the seasonal pattern in the posterior relative to the prior provides meaningful insight: the inversion's outcome is consistent with a chemically evolving atmosphere in which reduced $NO_x$ emissions contribute both directly and indirectly to observed $NO_2$ concentration trends.

**Specific Comments**

**Line 43:** Other methods of simultaneous lifetime/emission derivation have been demonstrated and evaluated with satellite measured $NO_2$ columns. A more comprehensive summary of the literature and prior analysis is needed here.

We added the below text to the introduction to better discuss other methods:

*"Research and refinement of inversion methods for estimating NOx emissions and lifetimes are crucial, especially for initiatives like the Copernicus CO2M mission (Sierk et al., 2021), which will utilize NO2 plumes to enhance CO2 monitoring by more accurately pinpointing emission sources. Several studies have quantified NOx emissions based on satellite NO2 retrievals by analyzing downwind plumes of NO2 from large sources, using inverse modeling computations with atmospheric chemical transport models (CTMs) (e.g., Brioude et al., 2013; Cheng et al., 2021; Kurokawa et al., 2009; Krol et al., 2024; Zhu et al., 2022). However, because CTMs can present accessibility challenges and require significant computational resources, alternative methods that do not rely on CTMs have been developed and applied to estimate NOx emissions and lifetimes (e.g., de Foy et al., 2014; Beirle et al., 2011)."*

**Line 115:** The ability of the superposition forward model to accurately represent the emissions/column relationship is tested in section 2.2. Photolysis representative for Riyadh is used in the MicroHH simulation, but the application city is Paris. This is confusing. Why was this choice made? Are there any city or latitude specific aspects of the model that are not directly transferrable and affect the interpretation?

This is a fair point. Ideally, the OSSE would have been conducted under conditions representative of Paris. We used the MicroHH simulation for Riyadh primarily for practical reasons: it was already available to us and provided a realistic, urban, and polluted environment to test the superposition model on synthetic data without requiring additional computationally expensive simulations.

This is also the first time the superposition model is tested against such high-resolution synthetic observations. Our goal here is to evaluate whether the model can reproduce the relationship between $NO_x$ emissions, lifetimes, and $NO_2$ columns in a controlled but realistic setting. For that purpose, we believe the Riyadh case is suitable.

We acknowledge that differences between Riyadh and Paris, such as photolysis rates, humidity, and emission characteristics (e.g. the VOC/$NO_x$ emission ratios), can affect atmospheric chemistry and thus the details of model performance. That said, the fundamental behavior of the model should remain applicable. We also note that the winter conditions in Riyadh (case 2) may resemble summer in Paris in terms of photochemical regime, suggesting some regime overlap and comparability. While even testing the OSSE in multiple cities would have been ideal, it was beyond the scope of this study.

**Figure 1:** Add description of black arrows; are these wind vectors at different locations over Paris?

The black arrows are indeed the wind vectors at different locations. We added the following line to the caption of figure 1: "The black arrows indicate the wind speed (from the north-east)"

**Line 124:** This implies that the symcity line densities are spaced by 5 km. In figure 3 c and f, The line densities are shown at a closer spacing of ~3 km.

We are glad that the reviewer spotted this. We indeed made the grid 10 times courser, so the initial MicroHH resolution was 300x300m and the coarsened one is 3x3km. We corrected this in the manuscript.

**Line 186:** "We use a prior lifetime uncertainty $s_{A,k}$ of 30%". This uncertainty is used for the inversion of the forward model described by equations 1 and 2, where k is the chemical loss rate constant in units of inverse time. With this wording and notation, it is unclear whether $s_{A,k}$ is referring to the uncertainty in the lifetime or in k. Since lifetime is the inverse of k, a 30% uncertainty in one value corresponds to a 233% uncertainty in the other. Further, the covariance in concentration and lifetime uncertainties is an element of the atmospheric chemistry. What are the downsides of not explicitly addressing this issue?

This is indeed the uncertainty in the lifetime of 30%, not in the decay rate. We corrected this throughout the manuscript to $\sigma_{A,\tau}$.

Our superposition model indeed treats lifetime as a separate parameter from emissions (and thus concentration), and assigns uncertainties to both independently. This simplifies the inversion but may neglect some nonlinearities or feedbacks that exist in the real atmosphere. Nevertheless, the simplification keeps our inversion framework manageable and remains appropriate for the spatial and temporal scales considered here. Also, the assigned uncertainties in emissions and lifetime implicitly capture part of this variability, and sensitivity tests confirm the robustness of our results to this assumption.

**Line 198:** Comment on how appropriate it is to treat the systematic uncertainty in the AMF error as random uncertainty used to draw from a normal distribution. At pixels with high emissions, the AMF error generally leads to VCDs that are biased low. This could lead to improper fitting of forward model parameters at those pixels.

Please see our response on the first major comment.

**Line 235:** Recommended to be reworded. To some readers, this may imply that the TROPOMI V2.4.0 product uses CAMS $NO_2$ profiles even though it uses profiles from TM5-MP at 1° x 1. Emphasize that the product used is the European product described in Douros et al. with 0.1° x 0.1° resolution profiles.

We agree and reworded this paragraph to: *"We use the European TROPOMI $NO_2$ product that uses CAMS a priori $NO_2$ profiles in the air mass factor and averaging kernel calculation. This product is based on the operational TROPOMI NO2 (v2.4.0) version, and is described in Douros et al. (2022)."*

**Line 238:** More context could be provided for the correction of TROPOMI bias. What was the existing TROPOMI bias at emission hotspots, and how much of this is corrected for with the 30% increase?

We included some more details on the TROPOMI bias, as described by Douros et al. (2022):

*"Douros et al. (2022) compared the standard and European TROPOMI NO$_2$ products with nine MAX-DOAS instruments, finding an average bias of –31% for the standard product and –19% for the European version. The European product reduces the bias by 5–18% at most stations and yields up to 30% higher NO$_2$ columns in emission hotspots, especially in winter. This supports its use for NO$_x$ emission estimates. We note that a bias is not necessarily a TROPOMI concern: a persistent low bias over urban areas may also stem from representativity differences: ground-based instruments sample narrow, localized air masses, while the TROPOMI pixel averages NO$_2$ over a larger and more heterogeneous area, often smoothing out urban pollution peaks."*

**Line 288:** See general comments; this is an area where more discussion of using CAMS OH for this purpose is warranted.

The CAMS 0.4° forecast product is part of the Copernicus Atmosphere Monitoring Service and provides global 5-day forecasts of atmospheric composition at approximately 0.4° (~40 km) resolution. It includes key chemical species such as NO$_x$, NO$_2$, O$_3$, CO, and OH, among others. Using CAMS OH and NO$_x$/NO$_2$ ratios as priors in our Bayesian inversion is justified by their physical coherence, spatiotemporal completeness, and compatibility with the scale of the rotated line densities and superposition column model, which requires one single effective NO$_x$ lifetime. We now include this discussion as requested.

**Line 293:** Expand on the justification of using a 30% uncertainty for the prior lifetime when the common value is 50%. The current explanation is that 30% encompasses most of the expected 50% uncertainty, but is the 50% uncertainty not also a type of standard deviation? If not, then clarify this.

The section that the reviewer is referring to is this: *"To incorporate the uncertainty in OH concentrations and its impact on the NOx lifetimes, we choose a standard deviation of 30% on the prior lifetime ($\sigma_{a,k}$) This selection aligns with the typical range of uncertainty observed in NOx lifetimes, which commonly falls within 50% (Lorente et al., 2019). By adopting a standard deviation of 30%, we encompass the majority of uncertainties within the expected 50% range, while also allowing for larger deviations in exceptional cases."*

We clarify why we choose the value of 30%: The previous study by Lorente et al. (2019) that we are referring to here, uses the 50% uncertainty on the NO$_x$ lifetimes as a strict cut off. In their method of fitting the superposition column model, the lifetime can not vary by more than 50%. In our study, we use a prior lifetime uncertainty ($\sigma_{a,k}$) of 30%, meaning that 95% (two standard deviations) of the lifetimes will fall within a 60% uncertainty, so our Bayesian cost function allows for slightly more deviation from the prior than in Lorente et al. (2019), as long as this improves the fit enough.

**Table 4 caption:** The standard deviation of the posterior is estimated using a specific date in the summer. Were other dates tested? The prescribed uncertainties may be expected to change throughout the year, such as during winter when the NO$_x$ lifetime is longer and its absolute uncertainty increases.

The reviewer raises a good point, and we agree that using only a single date is not very representative. As suggested, we repeated the same Monte Carlo test (50 runs) for 8 different dates throughout 2022. The results are shown in the table below.

| | | Stdev emissions (% deviation from mean) | Stdev lifetime (% deviation from mean) |
|---|---|---|---|
| Current | 16/06/2023 | 15% | 13% |
| Feb 2022 | 05/02/2022 | 2.0% | 1.9% |
| | 26/02/2022 | 0.3% | 1.6% |
| Apr 2022 | 11/04/2022 | 11% | 4.0% |
| | 22/04/2022 | 10% | 4.5% |
| Jul 2022 | 04/07/2022 | 2.9% | 2.3% |
| | 11/07/2022 | 3.9% | 2.3% |
| Oct 2022 | 04/10/2022 | 2.9% | 1.9% |
| | 25/10/2022 | 6.6% | 3.2% |
| **Average** | | **6.1%** | **3.9%** |

We do not observe a clear seasonal trend in the posterior error across these dates. This is likely because the prescribed uncertainties on the prior lifetimes and emissions (both set to 30%) are kept constant throughout the year in our main analysis, and we applied the same approach in this Monte Carlo sensitivity test.

To make the posterior error in our manuscript more representative, we have updated Table 4 in the manuscript to show the average posterior errors across these 8 days.

**Technical Corrections**

**Line 95:** Remove repeated "on"

This has been corrected.

**Line 115:** Change "Riaydh" to "Riyadh"

This has been corrected.

**Figure 3 caption:** Should be "symcity" instead of "simcity" for consistency?

This has been corrected.

**Table 1:** Add units to column "Total $E_{NOx}$"

Units have been added.

**Line 275:** Should this be "never completely *increased* to their original levels"?

Indeed, this has been corrected.

**References reviewer**

Beirle and T. Wagner, "A new method for estimating megacity $NO_x$ emissions and lifetimes from satellite observations," Atmospheric Meas. Tech., vol. 17, no. 11, pp. 3439–3453, Jun. 2024, doi: 10.5194/amt-17-3439-2024.

Benjamin de Foy, Joseph L. Wilkins, Zifeng Lu, David G. Streets, Bryan N. Duncan, Model evaluation of methods for estimating surface emissions and chemical lifetimes from satellite data, Atmospheric Environment, Volume 98, 2014,Pages 66-77, https://doi.org/10.1016/j.atmosenv.2014.08.051.

J.H. G. M. Van Geffen, H. J. Eskes, K. F. Boersma, and J. P. Veefkind, "TROPOMI ATBD of the total and tropospheric NO2 data products," no. 2.4.0, Jul. 2022, [Online]. Available: https://sentinel.esa.int/documents/247904/2476257/Sentinel-5P-TROPOMI-ATBD-NO2-data-products.pdf

J.L. Laughner, Zhu, Q., and Cohen, R. C., Evaluation of version 3.0B of the BEHR OMI $NO_2$ product, Atmos. Meas. Tech., 12, 129-146, https://doi.org/10.5194/amt-12-129-2019, 2019

J.L. Laughner, A.H. Zare and R.C. Cohen, Effects of daily meteorology on the interpretation of space-based remote sensing of $NO_2$ Atmos. Chem. Phys. 16, 15247-15264, doi:10.5194/acp-16-15247-2016, 2016.

J.L. Laughner and R. C. Cohen, "Direct observation of changing NOx lifetime in North American cities," Science, vol. 366, no. 6466, pp. 723–727, Nov. 2019, doi: 10.1126/science.aax6832.

Liu, F., Tao, Z., Beirle, S., Joiner, J., Yoshida, Y., Smith, S. J., Knowland, K. E., and Wagner, T.: A new method for inferring city emissions and lifetimes of nitrogen oxides from high-resolution nitrogen dioxide observations: a model study, Atmos. Chem. Phys., 22, 1333–1349, https://doi.org/10.5194/acp-22-1333-2022, 2022.

Jin, Q. Zhu, and R. C. Cohen, "Direct estimates of biomass burning $NO_x$ emissions and lifetimes using daily observations from TROPOMI," Atmospheric Chem. Phys., vol. 21, no. 20, pp. 15569–15587, Oct. 2021, doi: 10.5194/acp-21-15569-2021.

Zhu, J.L. Laughner, and R.C. Cohen, Estimate of OH Trends over One Decade in North American Cities, Proc. Nat. Acad. Sci. 10.1073/pnas.2117399119, 2022.

**References author reply**

Beirle, S., Boersma, K. F., Platt, U., Lawrence, M. G., & Wagner, T. (2011). Megacity emissions and lifetimes of nitrogen oxides probed from space. *Science, 333*(6050), 1737-1739.

Benjamin de Foy, Joseph L. Wilkins, Zifeng Lu, David G. Streets, Bryan N. Duncan, Model evaluation of methods for estimating surface emissions and chemical lifetimes from satellite data, Atmospheric Environment, Volume 98, 2014,Pages 66-77, https://doi.org/10.1016/j.atmosenv.2014.08.051.

Brioude, J., Angevine, W. M., Ahmadov, R., Kim, S. W., Evan, S., McKeen, S. A., ... & Trainer, M. (2013). Top-down estimate of surface flux in the Los Angeles Basin using a mesoscale

inverse modeling technique: assessing anthropogenic emissions of CO, NO x and CO 2 and their impacts. *Atmospheric Chemistry and Physics*, *13*(7), 3661-3677.

Cheng, X., Hao, Z., Zang, Z., Liu, Z., Xu, X., Wang, S., ... & Ma, X. (2021). A new inverse modeling approach for emission sources based on the DDM-3D and 3DVAR techniques: an application to air quality forecasts in the Beijing–Tianjin–Hebei region. *Atmospheric Chemistry and Physics*, *21*(18), 13747-13761.

Douros, J., Eskes, H., van Geffen, J., Boersma, K. F., Compernolle, S., Pinardi, G., ... & Veefkind, P. (2023). Comparing Sentinel-5P TROPOMI NO 2 column observations with the CAMS regional air quality ensemble. *Geoscientific Model Development*, *16*(2), 509-534.

Johnson, M. S., Philip, S., Meech, S., Kumar, R., Sorek-Hamer, M., Shiga, Y. P., & Jung, J. (2024). Insights into the long-term (2005–2021) spatiotemporal evolution of summer ozone production sensitivity in the Northern Hemisphere derived with the Ozone Monitoring Instrument (OMI). *Atmospheric Chemistry and Physics*, *24*(18), 10363-10384.

Krol, M., van Stratum, B., Anglou, I., & Boersma, K. F. (2024). Estimating NO x emissions of stack plumes using a high-resolution atmospheric chemistry model and satellite-derived NO 2 columns. *EGUsphere*, *2024*, 1-32.

Kurokawa, J. I., Yumimoto, K., Uno, I., & Ohara, T. (2009). Adjoint inverse modeling of NOx emissions over eastern China using satellite observations of NO2 vertical column densities. *Atmospheric Environment*, *43*(11), 1878-1887.

Lange, K., Richter, A., Schönhardt, A., Meier, A. C., Bösch, T., Seyler, A., ... & Burrows, J. P. (2023). Validation of Sentinel-5P TROPOMI tropospheric NO 2 products by comparison with NO 2 measurements from airborne imaging DOAS, ground-based stationary DOAS, and mobile car DOAS measurements during the S5P-VAL-DE-Ruhr campaign. *Atmospheric Measurement Techniques*, *16*(5), 1357-1389.

Liu, F., Tao, Z., Beirle, S., Joiner, J., Yoshida, Y., Smith, S. J., Knowland, K. E., and Wagner, T.: A new method for inferring city emissions and lifetimes of nitrogen oxides from high-resolution nitrogen dioxide observations: a model study, Atmos. Chem. Phys., 22, 1333–1349, https://doi.org/10.5194/acp-22-1333-2022, 2022.

Rijsdijk, P., Eskes, H., Dingemans, A., Boersma, K. F., Sekiya, T., Miyazaki, K., & Houweling, S. (2025). Quantifying uncertainties in satellite NO2 superobservations for data assimilation and model evaluation. *Geoscientific Model Development*, *18*(2), 483-509.

Sierk, B., Fernandez, V., Bézy, J. L., Meijer, Y., Durand, Y., Courrèges-Lacoste, G. B., ... & te Hennepe, F. (2021, June). The Copernicus CO2M mission for monitoring anthropogenic carbon dioxide emissions from space. In *International conference on space optics—ICSO 2020* (Vol. 11852, pp. 1563-1580). SPIE.

Tilstra, L. G., de Graaf, M., Trees, V., Litvinov, P., Dubovik, O., & Stammes, P. (2023). A directional surface reflectance climatology determined from TROPOMI observations. *Atmospheric Measurement Techniques Discussions*, *2023*, 1-29.

Zhu, J.L. Laughner, and R.C. Cohen, Estimate of OH Trends over One Decade in North American Cities, Proc. Nat. Acad. Sci. 10.1073/pnas.2117399119, 2022.

---

## Author Response (AR2)

**Author response to minor comments from Reviewer#2**

*Folkert Boersma, 20 October 2025*

We have addressed the minor comments by reviewer #2, and adapted the manuscript accordingly, as explained below in blue.

**Review**

This paper is interesting and should be pubblished.

We thank the referee for this assessment.

I realize on this reading that the analysis does not extend far enough from the peak of the plume to constrain the lifetime. I think this choice of downwind distance is what results in the poor characterization of lifetimes in this paper compared to others. A brief mention that other analyses try to follow the peak of the plume more than 1 e-fold down wind would be helpful in placing this paper in context of the analyses that have tried to get at NOx lifetime.

Our method does actually constrain the lifetime above the city. This is clearly demonstrated by the sensitivity study described in Section 3 which shows that the Bayesian inversion reproduces the domain-average lifetime in the MicroHH model - to within 35 minutes. We agree that including the decaying plume outside of the city in the study domain could possibly lead to stronger numerical constraints on the NOx-lifetime, but deliberately refrain from doing so, because of the strong differences in photochemistry between the urban area and the downwind plume, as shown in Figure 3. We now include the following explanation in section 3.2 that mentions this issue:

"Some studies estimate NO\x lifetimes by analyzing the exponential decay of the NO2 plume downwind of a city (e.g. Beirle et al. [2011], de Foy et al. [2014], Liu et al. [2022]). While this e-folding distance approach can provide additional constraints on the NOx lifetime compared to our method, which relies solely on the enhancement of NO2 over the city, it does not account for variations in photochemistry between the urban area and the downwind plume (as illustrated in Figure 3)."

Also, as a minor note, I expect formation of RONO2 as an NOx-sink—at a level of ~15%. Probably doesn't affect the results of this paper since the lifetime is interpreted as a regularization parameter and not chemically.

We agree that organic nitrate ($RONO_2$) formation can represent a non-negligible $NO_x$ sink, typically on the order of 10–20%. Our approach does not explicitly separate different chemical loss pathways, but fits an effective $NO_x$ lifetime (mentioned in sections 2 and 4.1.2) that implicitly includes all loss processes (e.g., $HNO_3$ and $RONO_2$ formation). We therefore expect that $RONO_2$ formation contributes to the effective loss rate retrieved, without affecting the conclusions of this study. We updated section 2 to

reflect this better, by mentioning that the main loss process is oxidation to HNO3, "with some loss to organic nitrates (RONO2)